# TimeOmni-1: Incentivizing Complex Reasoning with Time Series in Large Language Models

**Tong Guan**[1,2]   **Zijie Meng**[2]   **Dianqi Li**   **Shiyu Wang**   **Chao-Han Huck Yang**[3]
**Qingsong Wen**[4]   **Zuozhu Liu**[2]   **Sabato Marco Siniscalchi**[5,6]   **Ming Jin**[1*]   **Shirui Pan**[1*]

[1]Griffith University   [2]Zhejiang University   [3]NVIDIA   [4]Squirrel Ai Learning
[5]University of Palermo   [6]Norwegian University of Science and Technology

## Abstract

Recent advances in multimodal time series learning underscore a paradigm shift from analytics centered on basic patterns toward advanced time series understanding and reasoning. However, existing multimodal time series datasets mostly remain at the level of surface alignment and question answering, without reaching the depth of genuine reasoning. The absence of well-defined tasks that genuinely require time series reasoning, along with the scarcity of high-quality data, has limited progress in building practical time series reasoning models (TSRMs). To this end, we introduce **T**ime **S**eries **R**easoning **Suite** (TSR-Suite), which formalizes four atomic tasks that span three fundamental capabilities for reasoning with time series: (1) *perception*, acquired through scenario understanding and causality discovery; (2) *extrapolation*, realized via event-aware forecasting; and (3) *decision-making*, developed through deliberation over perception and extrapolation. TSR-Suite is the first comprehensive time series reasoning suite that supports not only thorough evaluation but also the data pipeline and training of TSRMs. It contains more than 23K samples, of which 2.3K are carefully curated through a human-guided hierarchical annotation process. Building on this foundation, we introduce **TimeOmni-1**, the first unified reasoning model designed to address diverse real-world problems demanding time series reasoning. The model is trained in multiple stages, integrating a mixture of task scenarios, novel reward functions, and tailored optimizations. Experiments show that TimeOmni-1 delivers strong out-of-distribution generalization across all tasks and achieves a high rate of valid responses. It significantly improves causality discovery accuracy (64.0% vs. 35.9% with GPT-4.1) and raises the valid response rate by over 6% compared to GPT-4.1 on the event-aware forecasting task. Code[1] and checkpoints[2] are publicly available.

## 1 Introduction

Time series data underpin a wide range of real-world systems, including energy, transportation, finance, and healthcare (Lu & Xu, 2024; Liu et al., 2023b; Guan et al., 2023; Lan et al., 2025). Comprehending real-world time series extends beyond mere pattern recognition, it necessitates multi-step and multi-hop reasoning to identify external factors driving temporal changes and to support downstream tasks that inherently build upon upstream pattern understanding and extrapolation (Kong et al., 2026). For instance, effectively scheduling energy demand requires integrating external knowledge such as extreme weather events, inferring causal mechanisms, anticipating event-driven variations, and ultimately supporting downstream decisions (MacKinlay, 1997; Liang et al., 2025). However, most existing time series approaches remain centered on basic pattern analytics and fall short in addressing such complex reasoning requirements, restricting their effectiveness in scenarios that demand a deeper understanding of context and robust decision-making support.

---

*Correspondence to: M. Jin <mingjinedu@gmail.com> and S. Pan <s.pan@griffith.edu.au>

[1] https://github.com/AntonGuan/TimeOmni-1
[2] https://huggingface.co/anton-hugging/TimeOmni-1-7B

Large language models (LLMs) have recently demonstrated impressive multi-step reasoning abilities across text, code, and mathematics (Wei et al., 2023; Shao et al., 2024). This potential for time series reasoning, however, remains largely untapped. The primary obstacle is the scarcity of large-scale multimodal time series alignment, instruction, and labeled chain-of-thoughts data during pretraining, which hinders the development of corresponding time series reasoning abilities. This capability gap is further evidenced on even leading LLMs (e.g., GPT-4.1) by recent benchmark (Merrill et al., 2024; Chow et al., 2024; Jin et al., 2024b). Furthermore, time series specific architectures such as Time-MoE (Shi et al., 2024) and Moirai (Woo et al., 2024) remain largely confined to forecasting tasks and lack the generalized reasoning capabilities required for broader applications. These gaps underscore the urgent need for dedicated time series reasoning models (TSRMs) that advance time series understanding, strengthen reasoning, and facilitate temporal analytics and knowledge generation, paving the way toward general-purpose time series intelligence.

However, two key limitations hinder the development of TSRMs: **(1) The scarcity of high-quality data to support general-purpose reasoning over time series.** Early efforts, such as constructing TSQA datasets (Kong et al., 2025), remain largely at the level of surface time series question answering and suffer from insufficient input context. Moreover, the formulation of time series reasoning tasks in existing multimodal datasets has not been systematically studied, leaving them unable to capture genuine reasoning depth with time series data. **(2) The lack of a validated and feasible pathway for effective time series reasoning across tasks.** It remains unclear which tasks genuinely demand reasoning capabilities over time series, as this question has not been systematically studied. This gap, combined with data scarcity, has confined existing research to narrow, task-specific settings. Many current approaches are trained independently for each task or even each dataset; for example, TimeMaster (Zhang et al., 2025b) employs six distinct models for six datasets. Such fragmentation hinders the transfer of reasoning capabilities across tasks and leaves the development of general-purpose time series reasoning an open challenge. These challenges naturally raise a pivotal question: How can we take a solid step toward fully *incentivizing reasoning capabilities in LLMs over time series*, so they can tackle complex real-world problems that inherently demand such reasoning?

Answering this question first drives us to tackle the challenge of data scarcity. Based on the limitations of existing time series QA datasets, we argue that time series reasoning tasks should adhere to two key principles. First, they should reward genuine reasoning rather than superficial pattern matching by systematically incorporating multi-step reasoning tasks and complete reasoning chains. Second, they should ensure context sufficiency to enable unambiguous answering or response generation, thereby strengthening the model's reasoning capacity and generalization across diverse scenarios. Guided by these principles, we formalize four atomic tasks that genuinely require reasoning with time series and introduce **TSR-SUITE**, which covers three fundamental time series reasoning capabilities: (1) *perception*, acquired through scenario understanding and causal discovery, reveals key temporal patterns; (2) *extrapolation*, realized via event-aware forecasting, predicts future trends and anomalies; and (3) *decision-making*, developed through perception and extrapolation, supports informed, adaptive actions. Building on this foundation, we present **TIMEOMNI-1**, the first generalized reasoning model for time series. The central premise is that effective time series reasoning requires internalizing fundamental temporal priors. To this end, TIMEOMNI-1 first injects the above three capabilities identified by TSR-SUITE into LLMs through supervised fine-tuning (SFT) as priors. We then design novel time series task-grounded rewards to cultivate genuine reasoning from these priors via policy optimization. Finally, to validate that these capabilities represent complementary facets of general time series reasoning, we unify all task capabilities within a single model with joint training.

Our contributions lie in three aspects:

1. **New Datasets and Testbed.** We introduce **TSR-SUITE**, the first comprehensive time series reasoning suite that formalizes four core tasks spanning three capabilities: perception, extrapolation, and decision-making. It contains more than 23K samples, of which 2.3K are carefully curated through a human-guided hierarchical annotation process. The suite serves not only as a testbed for thorough evaluation but also as a foundational data pipeline for training TSRMs.

2. **New Models.** We present **TIMEOMNI-1**, the first generalized reasoning model on time series data. It unifies diverse reasoning tasks within a two-stage curriculum: Stage 1 employs supervised fine-tuning with human-guided reasoning traces to inject temporal priors across the three key capabilities: perception, extrapolation, and decision-making; Stage 2 leverages reinforcement learning with novel task-grounded rewards to go beyond mimicking priors to robust reasoning.

3. **Comprehensive Evaluation and Key Insights.** TIMEOMNI-1 achieves Top-2 performance under both in-distribution (ID) and out-of-distribution (OOD) testbeds. Notably, it surpasses GPT-4.1 by 40.6% (ID) and 28.1% (OOD) in causal discovery accuracy while maintaining high valid-response rates across all tasks. Further experiments provide the first evidence that joint training across diverse time series reasoning tasks yields mutual gains across capabilities.

## 2 RELATED WORK

**Large Time Series Models.** Early efforts primarily aimed to endow time series models with zero-shot capability, mitigating domain-specific limitations when large-scale time series data were scarce. Time-LLM (Jin et al., 2024a) sought to transfer the generalization ability of LLMs into the time series domain. Prior developments of time series models are discussed in Appendix B. With the increasing availability of large-scale time series datasets, training **time series foundation models (TSFMs)** from scratch emerged as the mainstream approach (Woo et al., 2024; Ansari et al., 2024; Shi et al., 2024; Yao et al., 2025). These models demonstrated promising zero-shot performance but still fell short of supporting multi-task and multimodal capability. With the advent of multimodal models (Alayrac et al., 2022; OpenAI et al., 2024a) and reasoning-centric models (OpenAI et al., 2024b; Guo et al., 2025), the intersection of time series and LLMs has re-emerged at the forefront. Broadly, existing approaches can be categorized into two main groups. The first are **time series language models (TSLMs)**, which primarily adapt the language modeling paradigm to temporal data, focusing on supervised pattern fitting and QA without reasoning (Xie et al., 2025; Wang et al., 2025a; Kong et al., 2025; Zhang et al., 2025a; Yang et al., 2025; Wang et al., 2025c). The second are **time series reasoning models (TSRMs)**, distinguished by their attempt to employ reinforcement learning (RL) to cultivate genuine reasoning ability rather than pattern matching, and by their use of explicit reasoning to improve accuracy while providing interpretable, step-by-step explanations that enhance trustworthiness (Wang et al., 2025b; Tan et al., 2025b; Luo et al., 2025). However, this line of research is still in its infancy: the definition and necessity of reasoning remain vague, existing QA datasets are of limited quality, and most current works restrict themselves to single-task experiments under the R1-paradigm (Guo et al., 2025), without yet establishing a general-purpose reasoning framework for time series analogous to LLMs.

**Reasoning with Large Language Models.** Generative LLMs offer greater flexibility and generalization than traditional deep learning models, making them effective in complex tasks that require multi-step reasoning (Wei et al., 2023; Kojima et al., 2022). However, reasoning errors can propagate and degrade performance in some cases. To address this, Lightman et al. (2023) introduces a step-level reward mechanism into both data construction and model fine-tuning to enhance reasoning. DeepSeek-R1-Zero (Guo et al., 2025) shows that RL using only format and final answers can also improve reasoning. And it has expanded to math, code, translation, and multimodal tasks (Zhang et al., 2025d;c; Feng et al., 2025b; Huang et al., 2025; Feng et al., 2025a; Zhan et al., 2025; Liu et al., 2025). However, LLM-based reasoning for time series remains underexplored due to data scarcity.

## 3 METHODOLOGY

**Problem Definition.** We define time series reasoning as the process in which reasoning model (RM) $p_\theta$ first generate a sequence of intermediate rationales $R = (r_1, \ldots, r_K)$ and then produce a final answer $y$, conditioned on (1) observed time series inputs $X = \{x_{1:T}^{(m)}\}_{m=1}^{M}$ and (2) auxiliary context $C$ (e.g., task instructions or external knowledge). Formally:

$$(R, y) \sim p_\theta(R, y \mid X, C) = p_\theta(R \mid X, C)\, p_\theta(y \mid R, X, C). \tag{1}$$

This formulation covers both discrete-output tasks, where $y$ is a categorical option, and sequence-output tasks, where $y$ is a numerical sequence, under a unified reasoning framework. To standardize outputs, RMs must generate rationales $R$ within `<think></think>` tags, followed by the final answer $y$ enclosed in an `<answer></answer>` block. In contrast, non-reasoning models directly predict $y \sim p_\theta(y \mid X, C)$ without rationales, producing only the `<answer></answer>`.

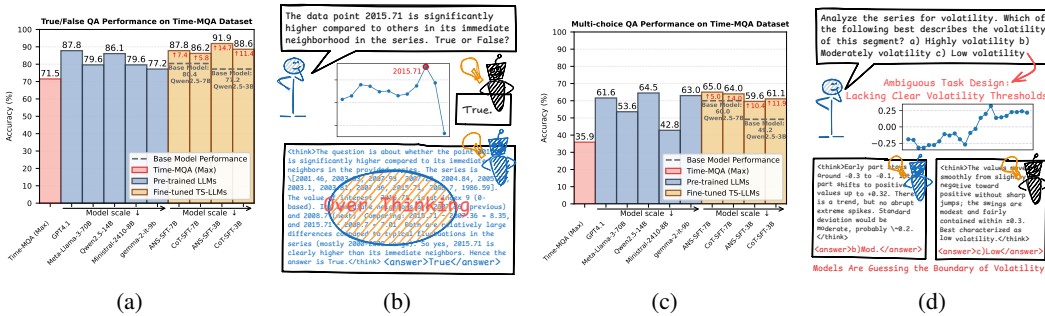

Figure 1: Limitations of existing TSQA dataset (Kong et al., 2025). **(a)** The marginal performance gap between RMs and NRMs. **(b)** Reasoning on simple TSQA leads to over-thinking. **(c)** Insufficient context leads to a performance plateau. **(d)** Ambiguous options forcing models to guess.

### 3.1 FORMULATING REASONING-CRITICAL TIME SERIES TASKS

**Limitations of Existing QA Tasks.** We use Time-MQA (Kong et al., 2025), the largest existing time series question answering (TSQA) dataset, to highlight two limitations. (1) *Many questions are overly simple and straightforward, where invoking reasoning leads to over-thinking.* At the aggregate level, as shown in Figure 1(a), the accuracy gap between stronger models (GPT-4.1) and smaller baselines (Qwen2.5-14B) is marginal, and in some cases reversed, indicating that additional reasoning capacity brings no benefit. Furthermore, all models achieve accuracy above 75%, highlighting that the tasks are not sufficiently challenging. At the instance level, as illustrated by the True/False QA from Time-MQA in Figure 1(b), the question can be directly answered by non-reasoning models, while reasoning complicates the process. (2) *Questions often lack sufficient input information, either in the time series $X$ or the context $C$, which prevents well-grounded answers and introduces ambiguity.* As shown in Figure 1(c), even advanced models plateau below 65% accuracy and show no gains after SFT. To investigate this, we conducted a human evaluation, which revealed numerous ambiguous cases caused by missing context. As exemplified in Figure 1(d), the options are not clearly distinguished (e.g., no explicit thresholds for high, moderate, and low volatility), forcing the model to guess rather than a reward modeling (Silver & Sutton, 2025) oriented reason. Consequently, errors reflect chance rather than insufficient time series reasoning ability (see Appendix C for detailed analysis of existing TSQA datasets). To address these issues, we propose two design principles for formulating time series QA tasks that require genuine reasoning.

**Principle 1 — QA-pairs must reward reasoning.** A reasoning model $M_{\text{RM}}$ explicitly generates rationales $R$ before producing the answer $y$, whereas a non-reasoning baseline $M_{\text{NRM}}$ directly outputs $y$. To determine whether a task requires reasoning, RMs should (significantly) outperform non-reasoning models (NRMs): $\bar{S}(M_{\text{RM}}) \gg \bar{S}(M_{\text{NRM}})$, where $\bar{S}(\cdot)$ denotes the mean score across tasks (e.g., accuracy for categorical prediction tasks or regression metrics for regression tasks).

**Principle 2 — QA-pairs must ensure context sufficiency.** Both the time series input $X$ and auxiliary context $C$ constitute the basis for reasoning. Unlike coding and mathematical problem solving, where a well-posed problem typically admits a unique solution (Imani et al., 2023), time series problem solving is especially sensitive to the sufficiency of $X$ and $C$. Let $K$ denote the number of ambiguous options. Even an ideal reasoner with infinite reasoning capacity ($RC \to \infty$) will be forced to guess if $X$ or $C$ is underspecified (e.g., missing thresholds for distinguishing high vs. low variance), while it should substantially exceed random guessing once $X$ and $C$ are sufficient:

$$\lim_{RC \to \infty} P(\text{correct} \mid X, C) \begin{cases} \approx \frac{1}{K}, & \text{if } X \text{ or } C \text{ is underspecified,} \\ \gg \frac{1}{K}, & \text{if } X \text{ and } C \text{ are sufficient.} \end{cases} \quad (2)$$

Therefore, ensuring context sufficiency is a critical design principle for formulating reasoning-critical time series tasks, as it prevents ambiguity and enables reasoning to be applied meaningfully.

**Reasoning-Critical Tasks.** The two principles motivate us to directly address the unique challenges of time series reasoning (ensuring QA requires reasoning and context sufficiency). To this end, we design a suite of tasks that form a progressive pathway covering three fundamental time series reasoning capabilities: (1) perception, (2) extrapolation, and (3) decision-making.

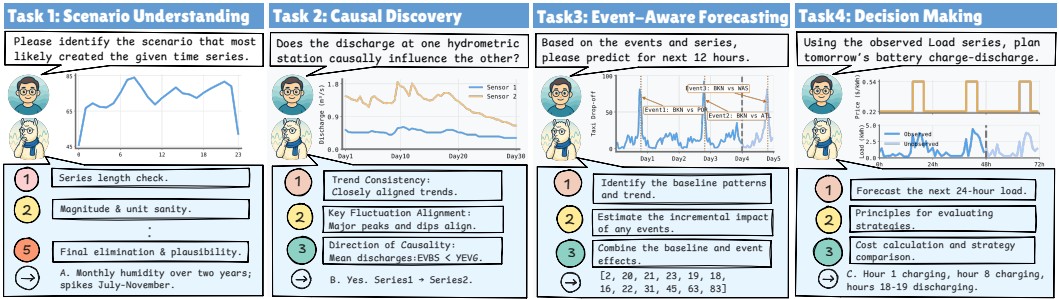

Figure 2: Illustrative examples of the four reasoning-critical time series tasks in TSR-SUITE.

As shown in Figure 2, the foundation of time series reasoning capabilities is perception, where the model first recognizes temporal patterns and then uncovers their underlying causes. This includes **Task 1: Scenario Understanding**, which focuses on single-series attribution by linking fluctuations to generative scenarios or external events (e.g., higher temperatures leading to increased ice-cream sales). It also encompasses **Task 2: Causality Discovery**, which extends attribution to the multi-series setting, requiring the model to compare trends across sequences and identify causal relations (e.g., upstream discharge influencing downstream flow). Together, these tasks ensure the model not only observes time series but also interprets them in a context-aware and causal manner. **Task 3: Event-aware Forecasting** requires the model to build on its perception ability to extrapolate future trajectories under explicit event perturbations. Accurate extrapolation depends on leveraging intrinsic temporal knowledge to analyze external events and infer their impact on temporal dynamics. Finally, **Task 4: Decision Making** represents the culmination of this chain. Building on the perception of temporal patterns (Task 1), causal relations (Task 2), and extrapolation (Task 3), the model must integrate these to select actions (Task 4) that maximize downstream utility (e.g., maximizing profits). By following the progressive capabilities of perception, extrapolation, and decision-making in formulating reasoning-critical tasks, we ensure that reasoning is an intrinsic requirement. Solving these tasks demands explicit reasoning from the outset, unlike conventional analytical tasks such as interpolation, where models often succeed through implicit fitting without reasoning.

## 3.2 TSR-SUITE

To mitigate the scarcity of data in the field, we construct **T**ime **S**eries **R**easoning **Suite** (TSR-SUITE), the first unified dataset suite tailored for time series reasoning. Unlike prior benchmarks designed purely for evaluation, TSR-SUITE is built as a training-and-evaluation suite that supports TSRMs development. The dataset spans 10 diverse domains and contains 23,605 curated QA pairs. Among them, 2,339 samples are annotated through a human-guided hierarchical annotation process. Detailed statistics for each task are provided in Appendix D.2. As shown in Figure 3(a), the data organization comprises three components as follows.

**Raw Data Collection.** Guided by the "perception–extrapolation–decision-making" pathway underlying our four tasks, we systematically collect publicly available time series data across 10 domains. Figure 3(a) provides an overview of the domain distribution, see Appendix D.1 for data source details.

**Task Formulation.** To align with our task design and support RL, we standardize the QA format across tasks. Specifically, Tasks 1, 2, and 4 are framed as discrete-output selection problems, while Task 3 is formulated as a sequence-output forecasting task, as shown in Figure 2. Each task adopts a customized data construction pipeline and is extensible to support further scaling with new input series. A key improvement over prior datasets is that our dataset, with over 23K QA pairs, is sufficiently large to support both training and evaluation (e.g., CiK with $355$ samples (Williams et al.), TSAIA with $1,054$ samples (Ye et al., 2025)), rather than serving solely as a testbed.

**Hierarchical Chain-of-Thoughts.** Existing time series QA datasets typically provide only labels (Kong et al., 2025), overlooking the fact that LLMs lack temporal priors for time series reasoning. To fill the gap, we design a hierarchical annotation pipeline involving an LLM Analyzer, Human Reviewers, and an LLM Rewriter (Figure 3(b)). **(1) Human-guided solvable annotation.** Instead of asking the LLM analyzer to directly solve the problems, we guide it with structured templates to elicit consistent reasoning, and we retain correctly solved samples as *Step-1 CoT data*. **(2) Context**

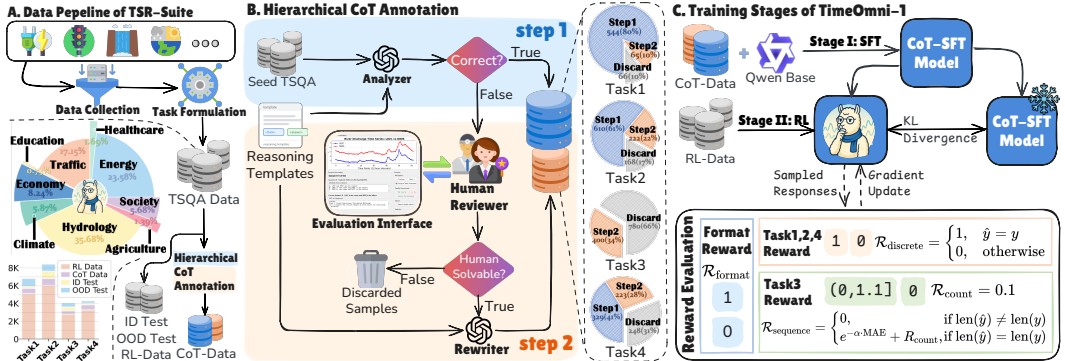

Figure 3: Overview of data and training pipeline. **(a)** Construction of TSR-SUITE, including *domain distribution* and *sample statistics*. **(b)** Hierarchical CoT annotation pipeline with outputs from each step for all tasks. **(c)** Two-stage training of TIMEOMNI-1: Stage 1 injects temporal priors via SFT; Stage 2 refines reasoning with task-grounded reward signals under RL.

**sufficiency verification.** For questions answered incorrectly in the first step, human experts use a customized evaluation interface (see Appendix D.5) to examine whether the provided context is sufficient to disambiguate the answer. If a question is solvable by human reviewers, expert-written reasoning chains are subsequently polished by the rewriter to follow our structured templates, and the resulting samples are collected as *Step-2 CoT data*. **Task 3 (Event-aware Forecasting) is treated as a special case**: unlike tasks with unique answers, forecasting outputs cannot perfectly match the ground truth due to inherent noise in real-world time series data. Human reviewers examine the cases and select 400 samples with relatively low mean absolute error (MAE). As a result, annotated predictions in Task 3 may not coincide exactly with the ground truth, but they capture plausible and well-justified reasoning. Additional analysis of Task 3 is provided in Appendix D.4.

### 3.3 TIMEOMNI-1

Developing time series reasoning poses unique challenges compared to other domains. Pretrained LLMs lack temporal priors, as they are not largely exposed to time series data during pretraining. To bridge this gap, we propose a two-stage training paradigm: (1) injecting temporal priors to anchor the model in a temporal knowledge space, and (2) refining these priors for robust reasoning through task-grounded rewards (Figure 3(c)). All experiments in this section use in-distribution (ID) testbeds.

**Stage 1: Injecting Time Series Reasoning Priors.** Human-guided reasoning priors instruct LLMs on how to decompose time series tasks into meaningful components. These traces narrow the exploration space to focus on three key capabilities (i.e., perception, extrapolation, and decision-making) instead of drifting toward commonsense heuristics or generic algebraic QA. We inject this knowledge through supervised fine-tuning (SFT). Implementation details of SFT are provided in Appendix F.1.

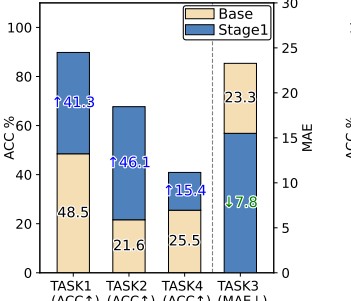

Figure 4: Stage 1 boosts accuracy; Base model at chance.

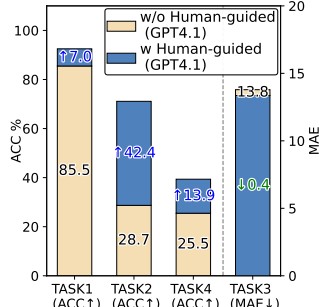

Figure 5: Human-guided templates are critical for priors.

**Finding 1:** Time series reasoning ability need not be innate; it can be effectively cultivated via supervised fine-tuning on a small set of high-quality, curated reasoning traces.

Base models without temporal priors collapse to chance-level accuracy when questions require fundamental temporal understanding (e.g., Task 2: 21.6% vs. 33.3% random guess in Figure 4).

Injecting reasoning traces, even with $<$1K seeds, boosts Task 2 accuracy by 46.1% after Stage 1, with comparable gains across other tasks. This prove that time series reasoning is not inherent to LLMs but can be systematically established through temporal priors.

**Finding 2:** Human-guided traces establish decomposition priors critical for time series reasoning.

Without guidance, LLMs tend to produce unstable, generic math-style reasoning traces that inconsistent across samples and fail to capture temporal dependencies. In contrast, when prompted with human-guided templates, the pretrained LLMs generates structured traces that explicitly follow decomposition strategies and achieve substantially higher accuracy. As shown in Figure 5, on GPT-4.1, human-guided templates improve zero-shot consistency accuracy from 28.7% to 71.1% on Task 2, with improvements also observed across all four tasks. These further confirm that pretrained LLMs lack temporal priors and must be enhanced through Stage 1 training.

**Stage 2: Refining Reasoning with Task-grounded Rewards.** While Stage 1 provides priors, they remain insufficient for robust reasoning. Stage 2 employs RL through group relative policy optimization (Shao et al., 2024) to turn mimicking priors into stable and generalizable reasoning behaviors (Figure 3(c)). Implementation details of RL stage are provided in Appendix F.2.

Here we focus on designing task-grounded, outcome-based rewards for time series reasoning, with detailed reward design provided in Appendix G. Each sample receives a reward composed of format verification and task correctness. $\mathcal{R}_{\text{format}}$ enforces the `<think></think><answer></answer>` schema. For correctness, we distinguish task types: for Tasks 1, 2, and 4, $\mathcal{R}_{\text{discrete}} \in \{0, 1\}$ denotes exact-match accuracy (1 if correct, 0 otherwise). For Task 3, we add a counting bonus $\mathcal{R}_{\text{count}} = 0.1$ if the predicted sequence length matches the required horizon. This structural reward is essential since LLMs still struggle with counting. For example, our Stage I checkpoint achieves only 55.7% success on sequence length. In addition, we use an exponential decay to map the unbounded MAE into a normalized range, which compresses arbitrarily large MAE toward zero to ensure higher rewards for smaller MAE.

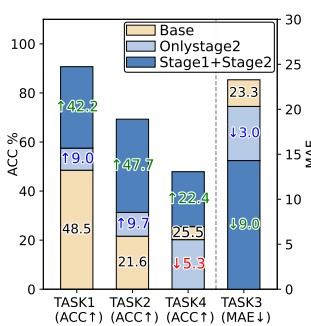

Figure 6: Stage 2 RL alone fails without Stage 1 priors.

**Finding 3:** Reinforcement learning works reliably only once the base model is anchored with fundamental temporal priors, which prevent collapse into spurious exploration.

Applying Stage 2 directly to a base model yields only marginal or even negative improvements (as shown in Figure 6, with a 5.3% drop on Task 4), since the rewards cannot distinguish genuine temporal knowledge from exploration within the pretraining corpus space. In contrast, when preceded by Stage 1, the same rewards refine temporal priors and progressively develop into robust reasoning.

**Joint Training for Time Series Reasoning.** Unlike prior single-task (or single-dataset) pipelines (Tan et al., 2025a;

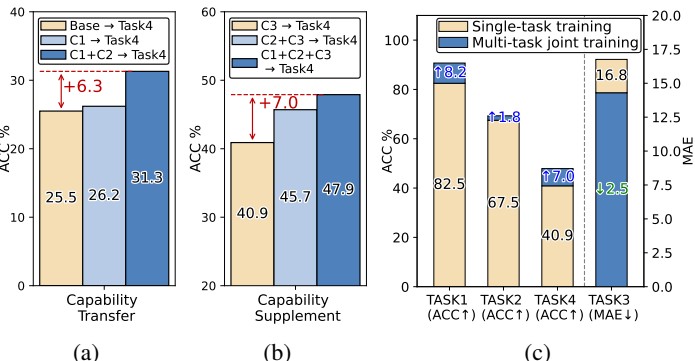

(a)      (b)      (c)

Figure 7: Joint training results: **(a)** Zero-shot capability transfer from precursor tasks (C1: Perception, tasks 1, task 2; C2: Extrapolation, task 3) to decision-making; **(b)** Supervised capability supplement by progressively incorporating tasks (C3: Decision-Making); **(c)** Joint training outperforms single-task training.

Zhang et al., 2025b), we investigate whether unifying perception, extrapolation, and decision-making objectives through joint training yields mutual benefits. We design the following two complementary experimental settings to systematically study the synergistic gains among the three reasoning capabilities.

> **Finding 4:** Joint training turns perception, extrapolation, and decision-making from silos into complementary capabilities, supporting a train-once, use-across-tasks paradigm for TSRMs.

*Progressive Capability Transfer.* This evaluates whether precursor reasoning capabilities transfer to downstream decision-making in a zero-shot manner. We evaluate three conditions on the ID decision-making testbed: (1) base model without precursor training, (2) model trained only on perception tasks, and (3) model trained on both perception and extrapolation tasks. As shown in Figure 7(a), accuracy on decision-making tasks increases from 25.5% to 26.2% and further to 31.3%, indicating that precursor capabilities enhance downstream reasoning even without direct supervision.

*Progressive Capability Supplement.* This assesses supervised joint training by gradually incorporating precursor tasks. We compare: (1) training solely on decision-making, (2) joint training on extrapolation and decision-making, and (3) full joint training across four tasks covers all three capabilities. Decision-making accuracy rises from 40.9% to 45.7% and peaks at 47.9%, as shown in Figure 7(b), confirming that progressively supplementing related tasks creates complementary learning benefits.

*Scaling to All Tasks.* Building on the above complementary settings, we compare single-task training against joint training across all four tasks. As shown in Figure 7(c), joint training consistently outperforms single-task training on the ID testbed. These results support a "train-once, use-across-task" paradigm for time series reasoning, where joint training effectively captures intrinsic connections within the temporal reasoning capabilities without task interference.

# 4 EXPERIMENTS

**Evaluation Metrics.** We observe that different models vary significantly in instruction-following ability, sometimes generating repetitive or malformed outputs. To ensure fair comparison, we adopt the standardized system prompt shown in Appendix E.2 and apply regular expressions to extract answers. We report the Success Rate (SR), which is the proportion of model outputs that yield a valid and extractable answer. All subsequent evaluation metrics are computed only on these valid cases, ensuring that performance reflects time series reasoning ability rather than instruction-following compliance. For discrete-output tasks (Scenario Understanding, Causality Discovery, Decision Making), we use Accuracy (ACC) via exact match. For the sequence-output task (Event-aware Forecasting), we use Mean Absolute Error (MAE) to assess forecasting precision. Higher ACC and lower MAE indicate better performance. The hyperparameters used are provided in Appendix H.

**Baselines.** We compare TIMEOMNI-1 against up-to-date models in two categories: **(1) Time series language models:** Time-R1 (Luo et al., 2025) (TSRMs for classical forecasting), Time-MQA (Kong et al., 2025) (fine-tuned 7B-8B models tailored for TSQA), and ChatTS (Xie et al., 2025) (fine-tuned

Table 1: Model comparison on four tasks, reporting ACC for Tasks 1, 2, and 4, MAE for Task 3, together with SR under ID and OOD testbeds. All metrics are computed only on valid responses. "–" denotes SR below 10%, where results are omitted for lack of statistical significance and therefore we choose not to report them. **Red**: the best, Blue: the 2nd best.

| Method | Base LLMs | Scenario Understanding | | | | Causality Discovery | | | | Event-aware Forecasting | | | | Decision Making | | | |
|---|---|---|---|---|---|---|---|---|---|---|---|---|---|---|---|---|---|
| | | ID | | OOD | | ID | | OOD | | ID | | OOD | | ID | | OOD | |
| | | ACC | SR% | ACC | SR% | ACC | SR% | ACC | SR% | MAE | SR% | MAE | SR% | ACC | SR% | ACC | SR% |
| **Proprietary Models** | | | | | | | | | | | | | | | | | |
| GPT-4.1-2025-04-14 | | 85.5 | 100.0 | **87.8** | 100.0 | 28.7 | 99.9 | 35.9 | 100.0 | **13.79** | 97.4 | 170.78 | 76.1 | 25.5 | 100.0 | 27.8 | 100.0 |
| GPT-4.1-Nano | | 66.2 | 97.5 | 62.6 | 98.7 | 29.8 | 98.6 | 28.0 | 98.4 | 18.98 | 92.8 | 170.78 | 76.1 | 28.9 | 99.5 | 34.1 | 97.8 |
| **Open-Source Models** | | | | | | | | | | | | | | | | | |
| Llama-3.1-70B-Instruct | | 56.4 | 100.0 | 59.6 | 100.0 | 23.4 | 100.0 | 28.9 | 99.9 | 24.67 | 92.8 | 238.98 | 97.0 | 20.3 | 96.8 | 17.7 | 97.4 |
| Mistral-Small-3.1-24B-Ins | | 64.8 | 100.0 | 69.2 | 100.0 | 24.6 | 100.0 | 25.8 | 100.0 | 17.28 | 72.0 | **120.97** | 43.4 | 44.7 | 100.0 | 42.9 | 100.0 |
| Llama-3.1-8B-Instruct | | 36.6 | 46.5 | 32.1 | 46.8 | – | 3.7 | – | 1.9 | 27.68 | 52.91 | 186.80 | 29.8 | 7.4 | 28.7 | 16.2 | 42.9 |
| Mistral-7B-v0.3 | | 40.5 | 92.2 | 34.7 | 87.6 | 29.0 | 86.0 | 26.9 | 82.6 | – | 5.3 | – | 0.0 | 24.3 | 94.2 | 16.7 | 96.7 |
| Qwen2.5-Instruct-7B | | 48.5 | 100.0 | 42.8 | 100.0 | 21.6 | 99.8 | 26.3 | 100.0 | 23.28 | 53.1 | 146.12 | 55.46 | 25.5 | 100.0 | 24.9 | 100.0 |
| **Time Series Language Models** | | | | | | | | | | | | | | | | | |
| Time-MQA | Llama3-8B | 32.2 | 29.5 | 25.1 | 32.6 | 30.1 | 44.3 | 31.2 | 37.2 | – | 1.4 | – | 0.4 | 12.0 | 13.3 | 11.6 | 15.8 |
| Time-MQA | Mistral-7B-v0.3 | 15.1 | 21.5 | 27.8 | 22.1 | 8.4 | 50.2 | 4.0 | 52.2 | – | 0.2 | – | 0.0 | 5.4 | 36.1 | 10.0 | 47.3 |
| Time-MQA | Qwen2.5-7B | 25.0 | 14.0 | 37.5 | 22.7 | 29.5 | 33.0 | 30.5 | 32.0 | 19.76 | 12.2 | – | 6.5 | 23.8 | 58.0 | 26.4 | 44.3 |
| ChatTS | | – | 6.0 | – | 6.9 | 18.2 | 30.1 | 18.6 | 26.7 | – | 0.0 | – | 0.0 | 5.8 | 27.1 | 11.1 | 27.1 |
| **Time Series Reasoning Models** | | | | | | | | | | | | | | | | | |
| Time-R1 | Qwen2.5-Instruct-7B | 30.9 | 94.0 | 34.0 | 92.5 | 30.2 | 53.8 | 31.4 | 48.9 | 17.61 | 38.7 | – | 6.3 | 27.8 | 95.7 | 32.2 | 93.1 |
| **Ours** | | | | | | | | | | | | | | | | | |
| TIMEOMNI-1 | Qwen2.5-Instruct-7B | **90.7** | 97.5 | 87.7 | 98.3 | **69.3** | 99.8 | **64.0** | 99.8 | 14.30 | 93.8 | 145.53 | 82.3 | **47.9** | 100 | **58.9** | 100 |

14B model for time series understanding). **(2) General-purpose LLMs:** two GPT-4.1 variants as proprietary representatives, and five open-source LLMs (7B to 70B) for comprehensive evaluation.

## 4.1 MAIN RESULTS

As shown in Table 1, **TIMEOMNI-1 consistently ranks among the top-2 models across all time series reasoning tasks.** Notably, it exceeds GPT-4.1 by **40.6%** (ID) and **28.1%** (OOD) on causal discovery. While achieving comparable accuracy on scenario understanding, TIMEOMNI-1 surpasses GPT-4.1 by a wide margin on tasks requiring deeper temporal priors (e.g., decision-making). Existing time series specialized models, however, exhibit weaker instruction-following ability than general LLMs (consistently lower SR). For example, ChatTS achieves 0% SR on the event-aware forecasting task; upon inspection, we found it fails to produce the required numeric sequences, generating only free-form text. This highlights a critical limitation of existing time series task-specific models: over-specialization compromises generalization ability.

## 4.2 MORE ANALYSIS

**General Reasoning Capability.** We evaluate whether our time series specialization diminishes general reasoning ability. We compare the base model, Stage 1 SFT model, and TIMEOMNI-1 on three general reasoning benchmarks: DROP (Dua et al., 2019), GPQA (Rein et al., 2024), and ReClor (Yu et al., 2020), which focus respectively on numerical reasoning, graduate-level knowledge reasoning, and logical reasoning. As shown in Figure 8, TIMEOMNI-1 improves average accuracy by 16.5% over the base model and 1.3% over the Stage 1 model. This indicates our approach not only maintains but also enhances general reasoning capabilities while specializing in time series tasks, avoiding the instruction-following degradation observed in other specialized models.

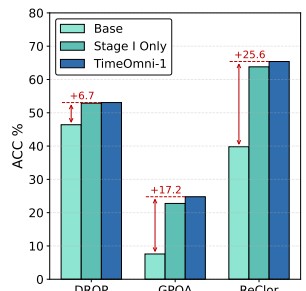

Figure 8: Performance on general reasoning benchmarks.

**Ablation on Training Stage.** We evaluate two configurations: (1) **Stage 1 models**, including NRMs via answer-only fine-tuning (ANS-SFT) and RMs via CoT fine-tuning (CoT-SFT); (2) **Stage 1+ Stage 2 models** (CoT-SFT+RL), which first activates reasoning via CoT-SFT and then applies RL. We analyze performance under multi-task joint training. As shown in Table 2, complete two-stage training (CoT-SFT+RL) delivers the most balanced performance, ranking Top-2. In causal discovery, CoT-SFT reaches 67.7% accuracy compared to only 30.5% for ANS-SFT, showing that answer-only supervision merely fits answer distributions without fostering reasoning. On decision-making, the CoT-SFT vs. ANS-SFT gap narrows from 10.1% (ID) to 5.5% (OOD), further confirming that ANS-SFT fails to foster reasoning, whereas CoT-SFT establishes transferable reasoning skills that are consolidated by RL in Stage II.

**Ablation on Training Strategy.** We compare single-task fine-tuning against multi-task joint training under identical training budgets. As shown in Table 2, multi-task joint training often enhances performance across all tasks and training stages. On the ID testbed, the jointly trained CoT-SFT+RL model (TIMEOMNI-1) achieves accuracy gains of 8.2%, 1.8%, 2.46 (MAE), and 7.0% across the four tasks compared to single-task training. Together with Figure 7, which demonstrates progressive capability *transfer* and *supplement*, these results validate that joint training effectively integrates temporal reasoning capabilities, reinforcing the "train-once, use-across-tasks" paradigm.

Table 2: Comparison across training stages and training strategies. **Red**: the best, Blue: the 2nd best.

| Training Stage | Training Strategy | Scenario Understanding | | | | Causality Discovery | | | | Event-aware Forecasting | | | | Decision Making | | | |
|---|---|---|---|---|---|---|---|---|---|---|---|---|---|---|---|---|---|
| | | ID | | OOD | | ID | | OOD | | ID | | OOD | | ID | | OOD | |
| | | ACC | SR% | ACC | SR% | ACC | SR% | ACC | SR% | MAE | SR% | MAE | SR% | ACC | SR% | ACC | SR% |
| **Base Model** | *Qwen2.5-Instruct-7B* | 48.5 | 100.0 | 42.8 | 100.0 | 21.6 | 99.8 | 26.3 | 100.0 | 23.28 | 53.1 | **146.12** | 55.5 | 25.5 | 100.0 | 24.9 | 100.0 |
| **ANS-SFT** | Single-task | 77.5 | 100.0 | 73.9 | 100.0 | 35.7 | 100.0 | 33.8 | 100.0 | 23.87 | 39.7 | 150.42 | 0.6 | 20.2 | 100.0 | 24.2 | 100.0 |
| | Multi-task Joint | 80.5 | 100.0 | 76.6 | 100.0 | 30.5 | 100.0 | 34.6 | 100.0 | **13.23** | 48.3 | 156.78 | 38.9 | **51.0** | 100.0 | 58.2 | 100.0 |
| **CoT-SFT** | Single-task | 73.9 | 100.0 | 85.6 | 83.9 | 66.3 | 96.0 | 61.8 | 92.4 | 15.10 | 64.6 | 157.21 | 34.5 | 39.4 | 98.40 | 47.3 | 94.87 |
| | Multi-task Joint | 89.8 | 88.0 | 85.8 | 88.2 | 67.7 | 98.8 | 61.1 | 97.0 | 15.53 | 71.8 | 151.41 | 55.7 | 40.9 | 100.0 | 52.7 | 100.0 |
| **CoT-SFT+RL** | Single-task | 82.5 | 100.0 | 87.0 | 98.7 | 67.5 | 99.6 | 61.7 | 99.0 | 16.76 | 79.2 | 169.88 | 66.0 | 40.9 | 100.0 | **63.6** | 99.6 |
| | Multi-task Joint | **90.7** | 97.5 | **87.7** | 98.3 | **69.3** | 99.8 | **64.0** | 99.8 | 14.30 | 93.8 | 145.53 | 82.3 | 47.9 | 100.0 | 58.9 | 100.0 |

## 5 CONCLUSION

In this paper, we introduce **TSR-SUITE**, which addresses the scarcity of reasoning-critical time series data. It formalizes four tasks across three fundamental capabilities for time series reasoning: perception, extrapolation, and decision-making. On this basis, we present **TIMEOMNI-1**, the first generalized, unified model for time series reasoning. It first injects temporal priors through supervised fine-tuning. Then, reinforcement learning with task-grounded rewards guides the model beyond mimicking priors toward robust reasoning. Experiments show that TIMEOMNI-1 achieves top-tier performance while preserving the general reasoning ability of the base model. Finally, we demonstrate that joint training across diverse reasoning tasks yields mutual gains, supporting a "train-once, use-across-tasks" paradigm for future time series reasoning models.

## ETHICS STATEMENT

Our work focuses solely on scientific challenges and does not involve human subjects, animals, or environmentally sensitive materials. We foresee no ethical risks or conflicts of interest. We are committed to upholding the highest standards of scientific integrity and ethical conduct to ensure the validity and reliability of our findings.

## REPRODUCIBILITY STATEMENT

To ensure reproducibility, we provide: Detailed hyperparameters for multiple training stages (Appendix H); All system prompts used in annotation, training, and evaluation (Appendix E). Our code and model checkpoints are publicly available.

## ACKNOWLEDGMENT

S. Pan was partially supported by Australian Research Council (ARC) under grants FT210100097 and DP240101547 and the CSIRO – National Science Foundation (US) AI Research Collaboration Program. This work was also supported by the NVIDIA Academic Grant in Higher Education and Developer program.

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

## A    THE USE OF LARGE LANGUAGE MODELS

During the preparation of this manuscript, we only employed large language models as auxiliary tools for non-substantive tasks. Their applications were limited to assisting in code debugging, checking grammar and formatting consistency, and improving the fluency of written text. The research design, experimental analysis, and conceptual contribution were independent of the LLMs output. All scientific insights and conclusions presented in this work are solely attributable to the authors.

## B    FURTHER RELATED WORK

Time series analysis has underpinned applications in finance, energy, transportation, healthcare, among others, over the past decade (Lu & Xu, 2024; Liu et al., 2023b; Guan et al., 2023; Lan et al., 2025). Most existing studies still concentrate on a single specific task, such as forecasting, classification, anomaly detection, and imputation (Guan et al.; Zhao et al.; Zheng et al.; Jiaheng et al.). These systems are typically **task-specific** and lack generality across tasks. Attempts such as TimesNet (Wu et al., 2023) and UNITS (Gao et al.) replace the output layer and loss to reuse a common backbone across tasks, but the resulting models still exhibit limited out-of-distribution (OOD) robustness. Transfer-learning approaches (Ismail Fawaz et al.; Wang et al.) and pre-trained models (Li et al., 2023) seek to mitigate OOD shifts; however, empirical evaluations typically remain **domain-specific** (e.g., different districts of one city) rather than achieving genuine transfer from domain A to domain B. A growing line of work further argues that a core bottleneck in time series analytics lies in the lack of integration with supplementary textual knowledge (Williams et al.; Ge et al., 2025), yet current models remain **modality-locked**, being unable to ingest such event information in textual form. In addition, most existing models adopt fixed output formats and depend on black-box computation, providing **limited interpretability**, even though some efforts rely on attention map (Liu et al., 2024), causal inference (Wang et al., 2024a), or visualization of hidden representations (Wang et al., 2024b; Cai et al., 2024a; Yi et al., 2024) to offer implicit explanations. Such latent interpretability, however, is often difficult for non-experts to understand or trust.

In contrast to these models, TSR-SUITE mitigates domain-specific brittleness by curating cross-domain, reasoning-critical time series data and expanding the task space beyond surface QA. And TIMEOMNI-1 improves reasoning accuracy and OOD generalization at scale across diverse domains through multi-task joint training. Finally, TIMEOMNI-1 yields step-by-step rationales that decompose temporal priors, event effects, and decision criteria, turning black-box predictions into transparent, reproducible reasoning.

## C    LIMITATIONS OF EXISTING TIME SERIES REASONING DATASETS

As a supplement to the Section 3.1, we also conducted additional experiments on existing datasets to assess how these limitations affect reasoning performance (shown in Figure 9): **(1) Reasoning necessity.** Across most tasks, Time-MQA, pre-trained LLMs, and fine-tuned TS-LLMs do not exhibit clear performance separation, indicating that the difficulty distribution lacks sufficient granularity to discriminate model capabilities. Moreover, CoT-based SFT does not outperform ANS-based SFT, suggesting that explicit reasoning is not required to solve these tasks. In fact, when tasks can be addressed through surface-level pattern matching, extra reasoning capacity yields only marginal gains. Therefore, we should construct dataset with calibrated difficulty gradients that genuinely require reasoning to solve. **(2) Context sufficiency.** For anomaly detection, current datasets lack relevant contextual information, forcing models to decide solely from the input sequences. As a result, both Time-MQA and pre-trained LLMs achieve only 50–60% accuracy, barely above random selection. After training, performance of ANS-based SFT improves by more than 20% on the ID subset, whereas CoT-based gains are less than half of that, indicating that improvements primarily come from directly mapping encoded sequence features to anomalous outcomes, rather than reasoning about the causes of anomalies. Consequently, for TSR with LLMs, we should prioritize providing sufficient context to support reasoning, rather than pushing models to overfit of time-series signals.

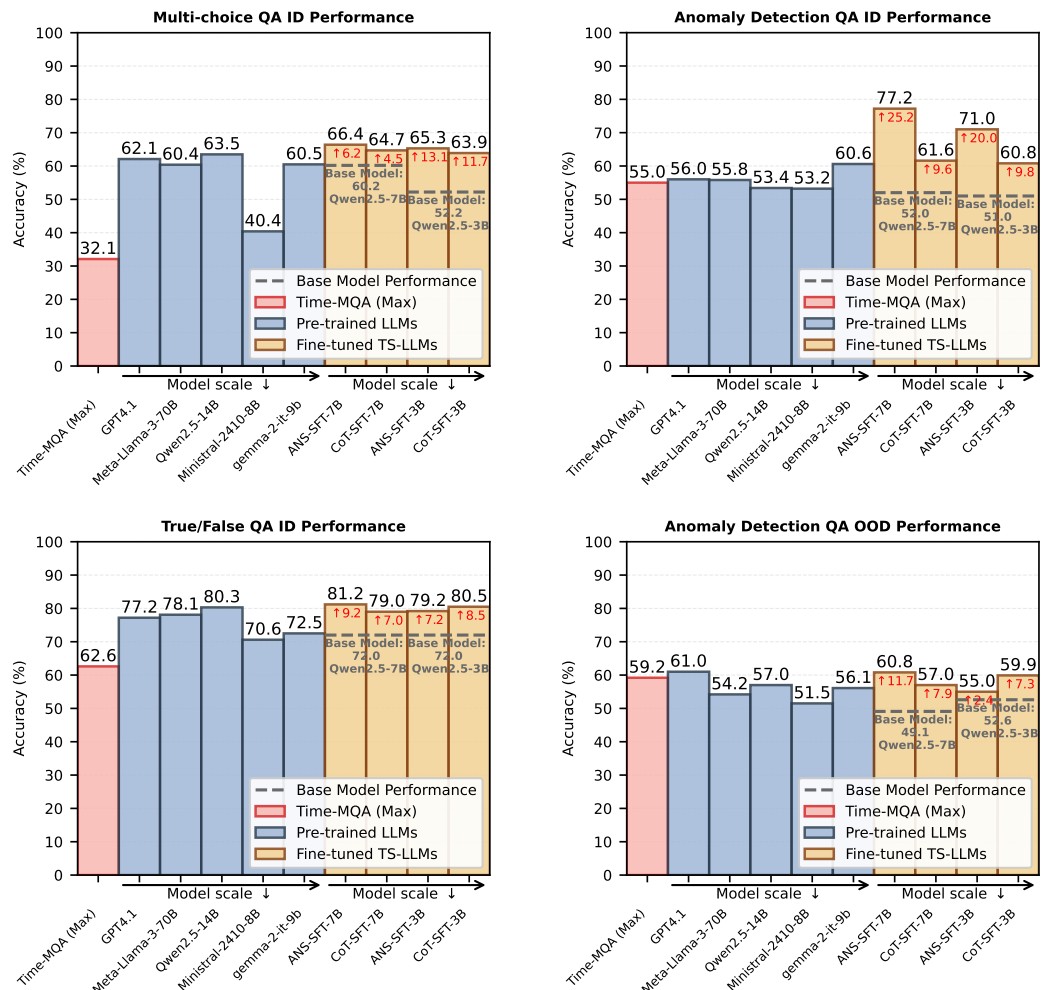

Figure 9: Evidence for the necessity of reasoning and the sufficiency of context on Time-MQA dataset: multiple-choice and true/false tasks saturate in in-distribution settings, while the anomaly-detection task exhibits apparent guessing under out-of-distribution shift.

# D DATASET DETAILS

## D.1 RAW DATA SOURCE

**Task 1: Scenario Understanding.** We collect diverse time series data across different scenarios from the work of Merrill et al. (2024). The out-of-distribution (OOD) test set is constructed based on the original domains of the time series data: specifically, samples from the *Agricultural*, *Education* and *Healthcare* domains are used as OOD test data.

**Task 2: Causality Discovery.** We perform causal discovery on river discharge time series from the CausalRivers dataset (Stein et al., 2025), aiming to uncover causal relationships from observational data. Ground-truth causal directions are determined according to river flow: the amount of water measured at an upstream station directly influences the amount measured downstream at a later time, and we thus consider such relations causal; if two rivers are not connected, we consider them non-causal. For the OOD test set, we split by geographical regions: training and in-distribution (ID) test data are taken from Eastern Germany, while the OOD test set is sourced from Bavaria.

**Task 3: Event-aware Forecasting.** We use a human mobility dataset as the primary training scenario, specifically taxi drop-off data near the Barclays Center in New York City. We collect the raw time series together with aligned events from Liang et al. (2024); Ge et al. (2025). For the OOD test set,

we adopt electricity load time series paired with weather events from the EWELD dataset (Liu et al., 2023a).

**Task 4: Decision Making.** To evaluate decision-making with counterfactual reasoning ability (i.e., reasoning about the outcomes of unobserved actions), we adopt a sandbox environment based on real building load data. Specifically, we use the CityLearn dataset (Nweye et al., 2023), which provides building load profiles and battery charge/discharge operations under a dynamic pricing scheme. Given 48 hours of historical building load and peak-valley pricing information, models are required to determine charge/discharge strategies for the next 24 hours. For the OOD test set, we select two buildings whose load patterns differ significantly from those in the training and ID test sets.

For all four tasks, we construct clear data pipelines, as detailed in Section 3.2, thereby facilitating future dataset expansions and task extensions.

## D.2  DATA STATISTICS

This section provides the detailed quantitative breakdown of TSR-SUITE, complementing the high-level overview in Section 3.2. Table 3 lists the number of samples available for each reasoning task, stratified by their use in the two-stage training (Stage 1 SFT and Stage 2 RL) as well as in-distribution (ID) and out-of-distribution (OOD) testbed. The data confirms a substantial scale for SFT (Stage 1) and an even larger set for RL (Stage 2), ensuring robust learning and generalization evaluation for each task.

| Task | #Stage 1 Train | #Stage 2 Train | #ID Test | #OOD Test |
|---|---|---|---|---|
| Scenario Understanding | 609 | 5104 | 200 | 899 |
| Causality Discovery | 778 | 6044 | 800 | 800 |
| Event-aware Forecasting | 400 | 2780 | 418 | 476 |
| Decision Making | 552 | 3284 | 188 | 273 |

Table 3: Detailed sample count statistics for the four time series reasoning tasks in TSR-SUITE across training stages and testbed.

## D.3  STATISTICS ON SEQUENCE LENGTH AND TOKEN BUDGET

In this section, we provide statistics for the actual sequence lengths used in TSR-SUITE in Table 4, as well as the corresponding token budgets computed using the tokenizer of our base model (Qwen2.5-Instruct-7B) in Table 5. These results clarify that our tasks involve substantially longer sequences than the illustrative examples in Figure 2. The average total token budget (1, 106 tokens) remains far below the maximum supported input length of Qwen2.5-Instruct-7B (32,768 tokens).

Table 4: Maximum and average time series lengths across four tasks.

| | Scenario Understanding | Causality Discovery | Event-aware Forecasting | Decision Making |
|---|---|---|---|---|
| **MAX / AVG length** | 800 / 316.3 | 124 / 121.7 | 96 / 78.3 | 48 / 48.0 |

Table 5: Average token budgets computed using the Qwen2.5-Instruct-7B tokenizer.

| | Scenario Understanding | Causality Discovery | Event-aware Forecasting | Decision Making | Overall |
|---|---|---|---|---|---|
| **AVG tokens of series $X$** | 1701 | 698 | 357 | 281 | 860 |
| **AVG tokens of context $C$** | 261 | 160 | 216 | 408 | 246 |
| **AVG total tokens** | 1962 | 858 | 573 | 689 | **1106** |

## D.4  TASK 3 (EVENT-AWARE FORECASTING) SPECIAL NOTES

Unlike multiple-choice tasks, where correct answers are explicitly listed among options, Task 3 requires forecasting future sequence within a fixed output window. This open-ended formulation

Table 6: MAE(↓) of CoT-SFT with different chain construction on Task 3.

|  | ID | OOD |
| --- | --- | --- |
| LLM Analyzer Generated | 15.10 | 157.2 |
| Ground Truth Guided | 24.53 | 395.56 |

significantly increases the difficulty of constructing coherent reasoning chains and prevents the LLM Analyzer from perfectly predicting results that are fully aligned with the ground truth. However, instead of guiding the generation of reasoning chains using the ground truth, we allowed the LLM Analyzer to generate predictions based on its own understanding. In our experiments, reasoning chains generated with ground-truth hints consistently resulted in worse CoT based SFT performance compared to those produced directly by the LLM Analyzer as shown in Table 6. This result also aligns with recent findings (Zhao et al., 2025; Gao et al., 2025; Li et al., 2025), which suggests that the most effective training data are instances slightly beyond a model's current ability but not prohibitively difficult. Furthermore, ground-truth–guided chains tend to obscure the task's inherent difficulty and deviate from the base model's natural data distribution. Therefore, we examine the cases and select 400 samples with relatively low MAE chains generated by the LLM Analyzer without relying on ground-truth hints as supervision for Stage 1 training to balance difficulty and quality.

### D.5 HUMAN EVALUATION INTERFACE

As described in Figure 3, when the LLM Analyzer (GPT-4.1 in our case) fails to solve a sample in Step 1, the instance proceeds to Step 2. In this step, human reviewers use the interface shown in Figure 10 to examine whether the provided context is sufficient to disambiguate the answer. If the question is solvable by human reviewers, their reasoning chains are further polished by the LLM Rewriter to follow our structured templates, and the resulting samples are collected as Step 2 CoT data.

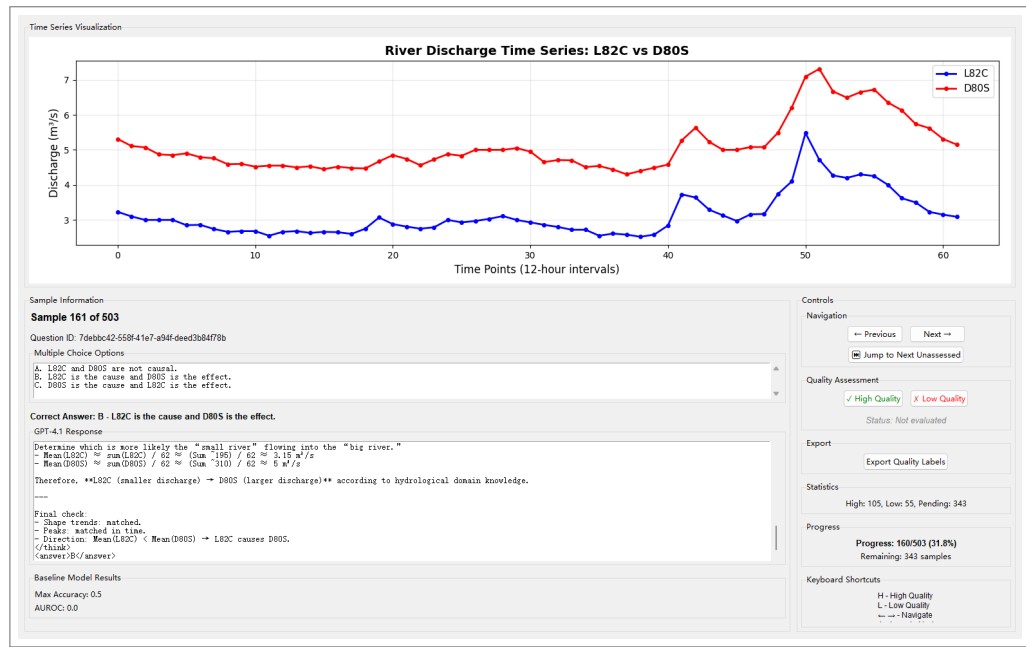

Figure 10: Screenshot of the human evaluation interface used in Step 2 of Hierarchical CoT Annotation.

# E PROMPT USED IN THIS PAPER

## E.1 HUMAN-GUIDED REASONING TEMPLATE FOR HIERARCHICAL CoT ANNOTATION.

As detailed in Section 3.2, our hierarchical annotation pipeline relies on structured reasoning templates to ensure consistency and quality in the Chain-of-Thought (CoT) generation process. These templates serve as explicit guidelines for the LLM Analyzer in the initial solving phase, as well as for human experts during verification and the LLM Rewriter in the refinement phase. The templates defined in this section are specifically designed for CoT annotation only. They provide a systematic framework for breaking down each reasoning task into logical steps, ensuring that all annotated traces follow a consistent structure while capturing the essential temporal reasoning processes. This approach guarantees that the resulting CoT data maintains high quality and facilitates effective model learning.

---

**Human-guided Reasoning Template for Task1: Scenario Understanding**

**Step 1. Series length check**
Observed length={L}. Expected per option: A{exp_len_A}; B{exp_len_B}; C{exp_len_C}; D{exp_len_D}. Retain option(s) whose expected length ≈ L.

**Step 2. Magnitude & unit sanity**
Value range={min}–{max}. Typical ranges: A{range_A}; B{range_B}; C{range_C}; D{range_D}. Eliminate options whose units/ranges mismatch.

**Step 3. Shape & temporal pattern**
Note trends/seasonality/spikes: {key_patterns}. Compare to option narratives: A{match_or_not}; B{match_or_not}; C{match_or_not}; D{match_or_not}. Keep best-matching narratives.

**Step 4. External-event alignment**
Identify clear events (e.g., single-day surge, mid-series drop): {events}. Which option explicitly explains this?

**Step 5. Final elimination & plausibility**
Remaining candidates: {remaining}. Choose the scenario that satisfies all of length, magnitude, pattern, and event consistency.

**Step 6. Double-check length consistency**
Confirm {tentative_choice} expected length =={L}? → {yes/no}. If "no", revert to next best candidate; else accept. Final choice: {chosen_option}.

---

**Human-guided Reasoning Template for Task3: Event-aware Forecasting**

**Step 1. Identify the baseline patterns**
You should first identify the baseline patterns and trend from the historical series.

**Step 2. Estimate the impact of any events**
Next, estimate the incremental impact of any special events (pre-event buildup, during-event lift, post-event dispersal) as an overlay on the baseline.

**Step 3. Combine the baseline and event effects**
Finally, given the current context, combine the baseline and event effects to generate the forecast sequence.

---

**Human-guided Reasoning Template for Task2: Causality Discovery**

**Step 1. Trend Consistency**
Check whether the two series demonstrate structurally consistent trends, such as shared "stable → rise → fall" shapes, both rise/fall at similar points (within ±1–2 time steps), and flat/stable periods aligned in time. It's OK for their absolute values to differ — match shape, not magnitude.
Red Flag: If one rises while the other stays flat or falls → Stop in the option: are not causal.
Be perceptually flexible. Flatness doesn't require perfect constancy — as long as fluctuations are very small relative to the scale of the full time series, they can still be considered flat.

**Step 2. Key Fluctuation Alignment**
Check whether the two time series have notable peaks, dips, or inflection points at the same or nearly the same time.
You must ensure: Spikes/dips occur within ±1–2 steps (which means ≤24 hours lag if sampling is 12h). For time series with low overall discharge (maybe near 0), even modest changes can be meaningful if they represent a clear pattern change relative to baseline. If peaks differ by ≥3 steps, it's too much lag to infer causality → the two time series are not causal. Do not confuse visually similar shapes with causality if key changes happen at clearly different times.

**Step 3. Direction of Causality**
Only perform this step if both Step 1 and 2 pass. Use the domain principle: "Small rivers flow into big rivers" — not the reverse.
Rule: If 'mean(A) < mean(B)', then 'A → B'. If 'mean(B) < mean(A)', then 'B → A'.
Notes: If Step 1 or 2 fails, skip Step 3. Timing matters: 3 steps of lag (36h) is already too much. Matching is about structure and timing, not numbers.

---

**Human-guided Reasoning Template for Task4: Decision Making**

**Step 1. Forecast the next 24-hour load**
Use the historical 48-hour load pattern to generate a forecast for tomorrow's 24-hour load. Pay special attention to the peak-price hours and estimate the likely loads during those hours.

**Step 2. Principles for evaluating strategies**
Charging should take place during off-peak hours when electricity price is low. Discharging should take place during peak hours when electricity price is high and forecasted load is significant. Avoid charging during peak hours or discharging during off-peak hours, as these operations increase cost instead of saving it.

**Step 3. Cost calculation and strategy comparison**
For each strategy, compute the expected saving using:

$$\text{Saving} = \sum_{h \in \text{peak}} \min\big(\hat{L}(h), P_{\max}^{\text{dis}}\big) \cdot (p_{\text{peak}} - p_{\text{valley}})$$

where $\hat{L}(h)$ is the forecasted load at hour $h$, $P_{\max}^{\text{dis}}$ is the maximum discharging power, and $p_{\text{peak}}$, $p_{\text{valley}}$ are the peak and valley electricity prices. Select the strategy with the highest saving.

---

E.2    SYSTEM PROMPT FOR TRAINING AND EVALUATION

This section presents the system prompts used in the ablation study on training stages (Section 4.2). The prompts are categorized into two types: **Chain-of-Thought(CoT)** prompts that require models to generate reasoning traces before answers, and **Answer-only(ANS)** prompts that directly output final answers without explicit reasoning.

For Tasks 1, 2, and 4, the CoT prompts enforce a structured output format where models must provide step-by-step reasoning within `<think>` tags before the final answer in `<answer>` tags. The ANS prompts for these tasks skip the reasoning step and output only the final answer. For Task 3, the prompts are adapted to accommodate sequence predictions while maintaining the same CoT/ANS

distinction. These prompts ensure consistent evaluation across different training configurations: ANS-SFT uses ANS prompts, CoT-SFT uses CoT prompts, and CoT-SFT+RL uses CoT prompts during both training stages.

---

**System Prompt of CoT for Task1, 2 and 4**

Output Format:
`<think>`Your step-by-step reasoning process that justifies your answer`</think>`
`<answer>`Your final answer(Note: Only output a single uppercase letter of the correct option)`</answer>`

---

**System Prompt of CoT for Task3**

You should think the impact of the event first, then output the predicted sequence.
Output Format:
`<think>`Your step-by-step reasoning process`</think>`
`<answer>`[Your predicted sequence]`</answer>`

---

**System Prompt of ANS for Task1, 2 and 4**

Output Format:
`<answer>`Your final answer(Note: Only output a single uppercase letter of the correct option)`</answer>`

---

**System Prompt of ANS for Task3**

You should output the predicted sequence directly.
Output Format:
`<answer>`[Your predicted sequence]`</answer>`

---

### E.3 MODEL ROBUSTNESS TO PROMPT VARIATIONS

In this section, we investigate the robustness of TIMEOMNI-1 to prompt variations. We design three prompt perturbations:

- **Paraphrased Question.** We use ChatGPT to rewrite the original question while keeping the semantic meaning unchanged.

- **Paraphrased System Prompt.** We use ChatGPT to paraphrase the system prompt while keeping the semantic meaning unchanged.

- **w/o System Prompt.** We entirely remove the system prompt to simulate the extreme case.

The experimental results are summarized in Table 7. The results show that paraphrasing either the question or the system prompt leads to minimal performance degradation, indicating that the model does not rely on specific wording or phrasing.

Surprisingly, even in the absence of the system prompt, the model remains highly robust. Although it no longer outputs the explicit `<think>` and `<answer>` tags, we adjust the evaluation script to parse the free-form responses and observe that TIMEOMNI-1 still maintains strong performance, sometimes even outperforming the original prompt. Manual inspection further reveals that the model continues to follow the reasoning template injected during Stage 1 training, suggesting that the temporal prior has been deeply internalized.

Overall, these results demonstrate that TIMEOMNI-1 is highly robust to prompt variations and resistant to perturbed instructions.

Table 7: Evaluation of TIMEOMNI-1 under prompt variations. **Red**: the best, Blue: the 2nd best. "–" denotes the **w/o System Prompt** setting, in which the model no longer produces the explicit `<think>` and `<answer>` tags; therefore we do not report SR for this condition.

| Prompt Variations | Scenario Understanding | | | | Causality Discovery | | | | Event-aware Forecasting | | | | Decision Making | | | |
|---|---|---|---|---|---|---|---|---|---|---|---|---|---|---|---|---|
| | ID | | OOD | | ID | | OOD | | ID | | OOD | | ID | | OOD | |
| | ACC | SR% | ACC | SR% | ACC | SR% | ACC | SR% | MAE | SR% | MAE | SR% | ACC | SR% | ACC | SR% |
| Paraphrase Question | 86.2 | 97.8 | 85.3 | 97.5 | 65.8 | 99.5 | 60.2 | 98.8 | **13.90** | 90.1 | 150.9 | 81.5 | 47.1 | 99.8 | 56.1 | 99.5 |
| Paraphrase System Prompt | 88.9 | 99.5 | 87.4 | 98.6 | 66.8 | 99.8 | 62.8 | 99.5 | 15.28 | 93.3 | 152.1 | 84.0 | 43.6 | 100.0 | 61.9 | 100.0 |
| w/o System Prompt | 86.5 | – | 87.1 | – | 68.5 | – | 63.3 | – | 14.29 | – | 150.1 | – | 40.4 | – | **63.0** | – |
| **Original Prompt** | **90.7** | 97.5 | **87.7** | 98.3 | **69.3** | 99.8 | **64.0** | 99.8 | 14.30 | 93.8 | **145.5** | 82.3 | **47.9** | 100.0 | 58.9 | 100.0 |

## F IMPLEMENTATION DETAILS OF THE TRAINING STAGES

Since there currently lacks a pre-trained time series encoder analogous to Vision Transformers (ViT) (Dosovitskiy et al., 2021) in the computer vision domain, we follow the common practice of tokenizing time series into text inputs, which aligns with the approaches adopted in Time-R1 (Luo et al., 2025) and Time-MQA (Kong et al., 2025).

### F.1 STAGE 1: SUPERVISED FINE-TUNING (SFT)

SFT is a process where a pre-trained model is further trained on a labeled dataset to adapt it for a specific task. This is achieved by minimizing the negative log-likelihood of the output given the input data. In the context of time series reasoning, the model learns to generate intermediate rationales and final answers based on observed time series data and auxiliary context. Specifically, based on a carefully curated dataset $\mathcal{D} = \{(X_i, C_i, R_i, y_i)\}_{i=1}^N$, the model's parameters $\theta$ are updated by minimizing the loss function:

$$\mathcal{L}(\theta; \mathcal{D}) = -\frac{1}{N} \sum_{i=1}^N \log \pi_\theta((R_i', y_i')|(X_i, C_i)) \tag{3}$$

where $R_i'$ and $y_i'$ are the predicted rationale and final answer, respectively. And for the $i$-th example, $X_i$ and $C_i$ represent the observed time series and auxiliary context.

### F.2 STAGE 2: REINFORCEMENT LEARNING (RL)

In reinforcement learning, we employ the group relative policy optimization (GRPO) algorithm (Shao et al., 2024) to refine our post-SFT model using carefully designed reward functions. Given an input pair $(X, C)$, GRPO samples $N$ rationale-answer trajectories $\{(R_i, y_i)\}_{i=1}^N$ from the policy model $\pi_\theta$, and organizes them into groups $\{\mathcal{G}_b\}_{b=1}^B$. The reward for trajectory $i \in \mathcal{G}_b$ is computed as

$$\hat{A}_i = \mathcal{R}(R_i, y_i) - \frac{1}{|\mathcal{G}_b|} \sum_{j \in \mathcal{G}_b} \mathcal{R}(R_j, y_j), \tag{4}$$

where $\mathcal{R}$ combines both the correctness of the answer and the quality of the response format. The policy is then updated using the following objective:

$$\mathcal{L}^{\text{GRPO}}(\theta) = \frac{1}{N} \sum_{i=1}^N \left[ \min \left( \frac{\pi_\theta(R_i, y_i \mid X, C)}{\pi_{\theta_{\text{refer}}}(R_i, y_i \mid X, C)} \hat{A}_i, \right. \right.$$
$$\left. \left. \text{clip} \left( \frac{\pi_\theta}{\pi_{\theta_{\text{refer}}}}, 1 - \epsilon, 1 + \epsilon \right) \hat{A}_i \right) - \beta \, D_{\text{KL}} \left( \pi_\theta \parallel \pi_{\theta_{\text{refer}}} \right) \right] \tag{5}$$

Here, $\pi_{\theta_{\text{refer}}}$ indicates the post-SFT model. $\epsilon$ and $\beta$ are hyperparameters that control the clipping threshold of the PPO update and the weight of the Kullback-Leibler (KL) divergence penalty, respectively.

## F.3 DISCUSSION OF TOKENIZING TIME SERIES INTO TEXT INPUTS

TIMEOMNI-1 employs a text-based strategy, representing time series as textual sequences. For a direct comparison, we also implement an embedding-based approach by training a time series encoder from scratch following the ChatTS (Xie et al., 2025) architecture, which is conceptually similar to OpenTSLM (Langer et al., 2025) and Time-LLM (Jin et al., 2024a). The same base model (Qwen2.5-Instruct-14B) is used in both settings to ensure a fair comparison. Results across all four tasks are summarized in Table 8, while inference efficiency under identical test conditions is provided in Table 9.

Our results indicate that a simple MLP-style encoder, similar to those used in ChatTS, OpenTSLM, and Time-LLM, does not provide a clear advantage over text serialization. Although the encoder improves Task 1 accuracy in-distribution, it leads to substantially lower success rates for Tasks 2 and 4, suggesting interference with the base model's instruction-following ability. A potential reason for the performance gap is the higher data requirement of the embedding-based method. Training a time series encoder from scratch is data-intensive, and our 2.3k CoT-Data might be inadequate.

The embedding-based design also slows inference by a factor of three because it introduces an additional neural path that cannot benefit from the kernel-level optimizations available in vLLM. For memory usage, the text-based approach requires a larger KV cache due to its longer tokenized representation; however, this overhead remains manageable and scales reasonably with input length. In our comparison in Table 9, the average time series length exceeds 300 time steps, yet the peak GPU memory remains below 52GB.

Taken together, the text-based strategy is both effective and efficient for the settings considered in this work, where sequence lengths and input dimensionality is modest. While dedicated time series encoders may offer advantages for higher-dimensional or extremely long sequences, the field currently lacks a widely adopted pretrained encoder analogous to ViT (Dosovitskiy et al., 2021) for vision. Developing such a general purpose time series encoder is an promising direction for our future work.

Table 8: Comparison of Text-based and Embedding-based Approaches. The same base model (Qwen2.5-Instruct-14B) is used in both settings to ensure a fair comparison. **Red**: the best, Blue: the 2nd best.

| Method | Scenario Understanding | | | | Causality Discovery | | | | Event-aware Forecasting | | | | Decision Making | | | |
|---|---|---|---|---|---|---|---|---|---|---|---|---|---|---|---|---|
| | ID | | OOD | | ID | | OOD | | ID | | OOD | | ID | | OOD | |
| | ACC | SR% | ACC | SR% | ACC | SR% | ACC | SR% | MAE | SR% | MAE | SR% | ACC | SR% | ACC | SR% |
| Base Model | 54.0 | 100.0 | 54.5 | 100.0 | 30.5 | 100.0 | 31.4 | 99.8 | 20.13 | 88.0 | **131.05** | 64.1 | 35.1 | 100.0 | 34.8 | 100.0 |
| Text-based (Ours) | 74.2 | 100.0 | 69.4 | 99.3 | **49.5** | 100.0 | **47.9** | 98.2 | **15.22** | 90.2 | 139.44 | 84.9 | **38.9** | 100.0 | **51.1** | 99.0 |
| Embedding-based | **88.5** | 92.9 | **82.4** | 97.7 | 40.3 | 68.7 | 38.7 | 62.9 | 22.15 | 75.1 | 162.23 | 58.4 | 18.2 | 59.5 | 20.0 | 51.9 |

Table 9: Inference efficiency comparison.

| Method | Samples | Avg Inference Speed per Sample (s) | Peak GPU Memory (GB) |
|---|---|---|---|
| Text-based (Ours) | 100 | 7.07 | 51.77 |
| Embedding-based | 100 | 22.01 | 32.04 |

## G  TIME SERIES TASK-GROUNDED REWARD DESIGN

All samples are required to follow a *basic format reward* $\mathcal{R}_{\text{format}}$, which checks whether outputs comply with the schema `<think>...</think><answer>...</answer>`. For discrete-output tasks (scenario understanding, causality discovery, decision-making), correctness is directly measurable by

$$\mathcal{R}_{\text{discrete}} = \begin{cases} 1, & \hat{y} = y, \\ 0, & \text{otherwise}. \end{cases} \quad (6)$$

For sequence-output task (event-aware forecasting), we define a continuous reward based on the exponential decay of mean absolute error (MAE):

$$\text{MAE} = \frac{1}{T} \sum_{t=1}^{T} |\hat{y}_t - y_t|, \quad (7)$$

$$\mathcal{R}_{\text{sequence}} = \begin{cases} 0, & \text{if } \text{len}(\hat{y}) \neq \text{len}(y), \\ \exp(-\alpha \cdot \text{MAE}) + \mathcal{R}_{\text{count}}, & \text{if } \text{len}(\hat{y}) = \text{len}(y), \end{cases} \quad (8)$$

together with a horizon-matching bonus $\mathcal{R}_{\text{count}} = 0.1$. And the final per-sample reward integrates all components as:

$$\mathcal{R}_i = \lambda \mathcal{R}_{\text{format}} + (1 - \lambda) \mathcal{R}_{\text{task}}, \quad (9)$$

where $\mathcal{R}_{\text{task}}$ refers to $\mathcal{R}_{\text{discrete}}$ for discrete-output tasks and $\mathcal{R}_{\text{sequence}}$ for sequence-output task. We set $\lambda = 0.1$ in all experiments.

### G.1  REWARD SENSITIVITY ANALYSIS

In this section, we conduct controlled sensitivity sweeps over the key reward components ($\alpha$, $R_{\text{count}}$, $\lambda$), and evaluate their effects on training stability. All experiments are trained for 1 epoch. For $\alpha$ and $R_{\text{count}}$, which only affect the reward design of Task 3, we report the MAE curves for Task 3 over 1 epoch. For $\lambda$, we evaluate the proportion of responses with a positive task reward $\mathcal{R}_{\text{task}}$ over 1 epoch, where $\mathcal{R}_{\text{task}}$ denotes $\mathcal{R}_{\text{discrete}}$ for discrete-output tasks and $\mathcal{R}_{\text{sequence}}$ for the sequence-output task.

**$\alpha$ and $R_{\text{count}}$ exhibit low sensitivity.** Varying these coefficients leads to only mild changes in MAE curves, indicating that the RL process remains stable under a broad range of settings. **Format reward is essential**. When disabling the format component ($\lambda = 0$), the model collapses after 70 steps, showing a sharp drop in the positive ratio of $\mathcal{R}_{\text{task}}$. This confirms that format consistency is necessary for stable RL optimization.

Overall, the current hyperparameter choices yield the better trade-off between stability and performance. These analyses provide clear guidance for reproducing and tuning the reward design.

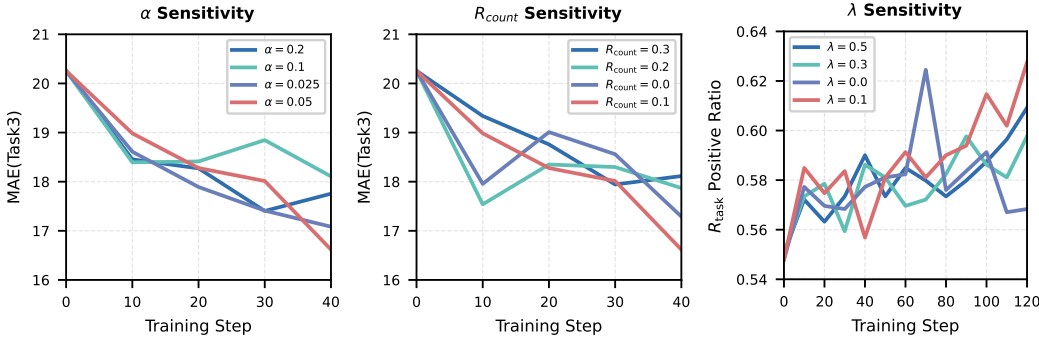

Figure 11: Reward sensitivity across $\alpha$, $R_{\text{count}}$, and $\lambda$. The model is robust to $\alpha$ and $R_{\text{count}}$, whereas disabling the format reward ($\lambda = 0$) causes training collapse.

## H    TRAINING CONFIGURATION

Our training process follows a two-stage procedure, consisting of supervised fine-tuning (SFT) followed by reinforcement learning (RL). For the SFT stage, we begin by fine-tuning the Qwen2.5-7B-Instruct (Team, 2024) with full-parameter updates for a total of 1 epoch, utilizing DeepSpeed ZeRO-3 (Rasley et al., 2020) for efficient training. The fine-tuning is performed in BF16 precision with FlashAttention-2 enabled to accelerate attention operations. The maximum sequence length is set to 8192, and the per-device batch size is 1, with gradient accumulation of 32. The optimization procedure uses a peak learning rate of $1.0 \times 10^{-5}$ with a cosine learning rate scheduler and the warm-up ratio as 0.1. All training is carried out using the LLaMA-Factory repository (Zheng et al., 2024) on a system equipped with a single NVIDIA H200-140G GPU. In the RL stage, we continue training from the Stage 1 checkpoint using the verl repository (Sheng et al., 2024) and FSDP (Zhao et al., 2023) under BF16 precision. The maximum sequence length is reduced to 2048, and training is performed across 8 NVIDIA A100-80G GPUs. The training batch size is 128, with RL minibatches of size 32 and per-GPU micro-batches of 8. Gradient clipping is applied with a maximum global norm of 3.0, and a KL-penalty coefficient of 0.04 is used to regulate the model. For rollouts, 8 trajectories are collected per update, with a sampling temperature of 0.7. The learning rate is set to $1.0 \times 10^{-6}$, and training continues for 3 epochs.

## I    REINFORCEMENT LEARNING TRAINING DYNAMICS

This section reports the full reinforcement learning (RL) training dynamics in Figure 12. The plots track both the overall reward and the format correct ratio for the training split and the OOD testbed throughout the RL stage. To reduce evaluation cost during RL, we cap the maximum response length at 2048 tokens during training. As a result, the performance in these curves may appear slightly lower than the full results reported in Table 1, which is expected. Overall, the curves provide a clear view of the optimization behavior and demonstrate that RL training for TIMEOMNI-1 remains stable and continues to improve over training steps.

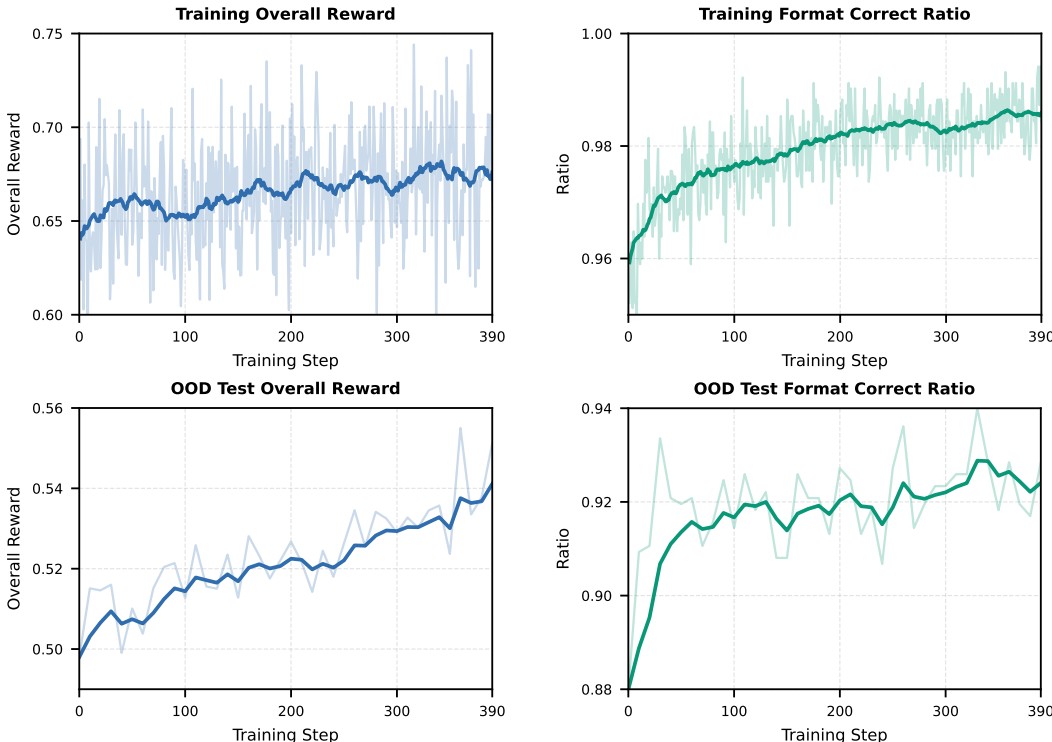

Figure 12: Full reinforcement learning training dynamics showing reward progression and format correctness for the training split and the OOD testbed.

# J    SCALING OF TRAINING DATASET

In this section, we analyze the scaling behavior of TIMEOMNI-1 by varying the amount of training data used in Stage 1 (SFT) and Stage 2 (RL). All experiments strictly follow the training configurations reported in Appendix H. All Stage 1 (SFT) data scaling runs are trained for 1 epoch, and all Stage 2 (RL) data scaling runs are trained for 3 epochs. Figures 13 present the scaling behavior of Stage 1 (SFT) as we vary the amount of CoT-Data (25%, 50%, 75%, 100%). Figures 14 present the scaling behavior of Stage 2 (RL) as we increase the amount of RL-Data (25%, 50%, 75%, 100%).

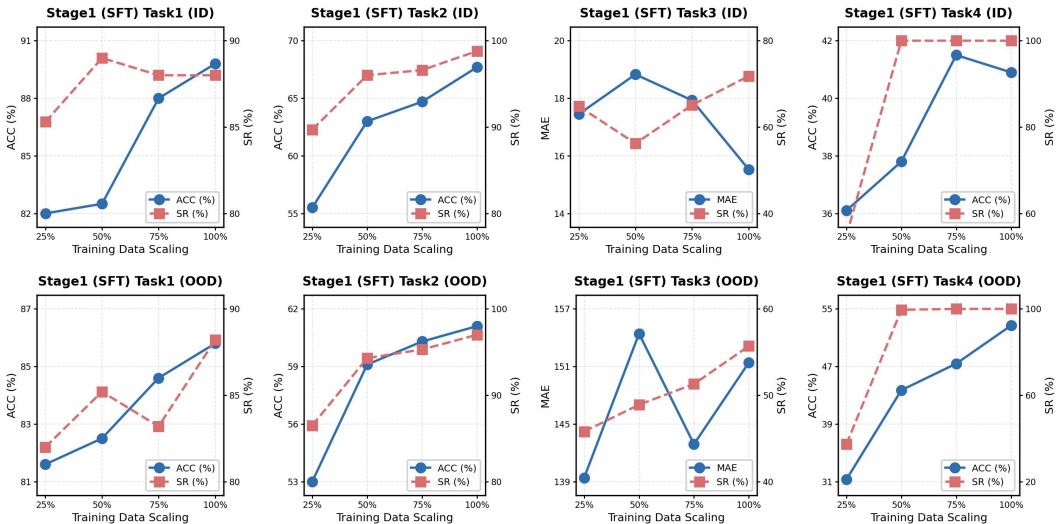

Figure 13: Scaling of Stage 1 (SFT) Training Dataset.

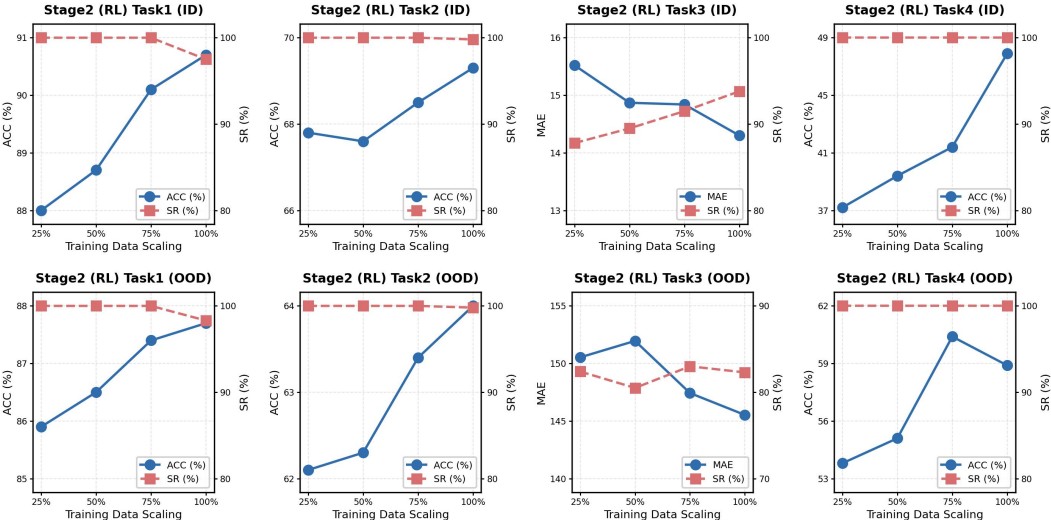

Figure 14: Scaling of Stage 2 (RL) Training Dataset.

Across all four tasks and both ID/OOD testbeds, the primary ACC/MAE metrics follow a clear scaling pattern: larger training sets monotonically improve Task 1, Task 2, Task 4 ACC and reduce Task 3 MAE. Notably, Task 3 MAE does not perfectly follow a scaling trend during SFT, which suggests that sample quality may influence the SFT phase more strongly. However, once the model enters Stage 2 where it performs its own exploration and refinement, Task 3 MAE follows a more clear scaling pattern and continues to decrease. In Stage 1, SR for Task 1, Task 2, Task 4 is already saturated near 100%. In Stage 2, RL continues to improve the SR for Task 3. Together, these curves show that both training stages benefit from increased data.

# K    ADDITIONAL EVALUATION ON EXTERNAL TIME SERIES QA BENCHMARKS

To further validate the generalization ability of TIMEOMNI-1, we conduct experiments on **three external time series QA benchmarks** spanning **13 tasks and 3,406 samples**:

- **MTBench** (Chen et al., 2025) (real-world time series; 4 tasks; 2,380 samples),
- **TimeSeriesExam** (Cai et al., 2024b) (synthetic time series; 5 tasks; 746 samples),
- **CaTS-Bench** (Zhou et al., 2025) (real-world time series; 4 tasks; 280 samples).

We evaluate TIMEOMNI-1 alongside two baselines: (1) **Qwen2.5-Instruct-7B**, the base model of TIMEOMNI-1, and (2) the **original results** reported in the original paper.

Table 10: **MTBench** results (real-world dataset). **Red**: the best, Blue: the 2nd best.

| Method | FinanceQA (7-Day) | WeatherQA (7-Day) | FinanceQA (30-Day) | WeatherQA (14-Day) | AVG |
|---|---|---|---|---|---|
| *Num. Samples* | 484 | 666 | 523 | 707 | – |
| **Base Model (Qwen2.5-Instruct-7B)** | 62.2 | 44.7 | 53.5 | 36.2 | 49.2 |
| **Original Paper (GPT-4o)** | 65.1 | 41.7 | 52.8 | **44.8** | 51.1 |
| **TIMEOMNI-1** | **72.1** | **51.4** | **66.2** | 42.6 | **58.1** |

Table 11: **TimeSeriesExam** results (synthetic dataset). **Red**: the best, Blue: the 2nd best.

| Method | Anomaly Detection | Causality | Noise Understanding | Pattern Recognition | Similarity Analysis | AVG |
|---|---|---|---|---|---|---|
| *Num. Samples* | 108 | 72 | 84 | 362 | 120 | – |
| **Base Model (Qwen2.5-Instruct-7B)** | 37.0 | 44.4 | 44.0 | 50.0 | **59.2** | 46.9 |
| **Original Paper (Phi-3.5)** | 28.0 | **52.0** | 26.0 | 47.0 | 45.0 | 39.6 |
| **TIMEOMNI-1** | **42.6** | 37.5 | **56.0** | **56.4** | 46.7 | **47.8** |

Table 12: **CaTS-Bench** results (real-world dataset). **Red**: the best, Blue: the 2nd best.

| Method | Caption Retrieval | TimeSeries Retrieval | Amplitude Comparison | Mean Comparison | AVG |
|---|---|---|---|---|---|
| *Num. Samples* | 100 | 100 | 40 | 40 | – |
| **Base Model (Qwen2.5-Instruct-7B)** | **69.0** | **46.0** | 87.5 | 57.5 | **65.0** |
| **Original Paper (LLaVA v1.6 Mistral)** | 44.0 | 29.0 | 43.0 | 35.0 | 37.8 |
| **TIMEOMNI-1** | 57.0 | 37.0 | **92.5** | **67.5** | 63.5 |

MTBench is a multimodal time series QA benchmark where the auxiliary text $C$ contains task-relevant information (e.g. external events) rather than only task instructions. This setting closely matches the formulation of TSR-SUITE, where the joint distribution

$$(R, y) \sim p_\theta(R, y \mid X, C) = p_\theta(R \mid X, C)\, p_\theta(y \mid R, X, C)$$

ensures that both the reasoning path $R$ and the final answer $y$ depend on the interaction between the time series input $X$ and the contextual information $C$. Owing to this strong alignment, TIMEOMNI-1 achieves the largest gains on MTBench, even surpassing the originally reported GPT-4o performance.

By contrast, the textual input in TimeSeriesExam and CaTS-Bench serves primarily as task instructions, meaning that the reasoning is dominated by the time series modality and is less aligned with the multimodal reasoning design of TSR-SUITE. Even under this more restricted setup, TIMEOMNI-1 outperforms the base model, demonstrating that it has acquired time series reasoning abilities that transfer robustly across domains and tasks.

## L CASE STUDY

Table 13: Task1 (Scenario Understanding) example with options and reasoning process.

**Task1: Scenario Understanding**

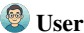 **User**

You are given a time series. Please identify the scenario that most likely created it.

[39, 58, 51, 45, 40, 39, 57, 45, 45, 34, 40, 32.64, 62, 92, 37, 31, 29.52, 46, 69, 79, 43, 51, 50, 50, 56, 45, 57, 32, 35, 55, 38, 41, 38, 54, 33, 48, 54, 40.45, 30.31, 56, 49, 37.41, 45, 39, 49, 31.52, 47, 51, 33, 53, 38.49, 33, 30, 35.52, 41, 32.61, 52, 44, 29.68, 41, 32.44, 46.5, 49, 48, 38, 46, 38, 49, 45, 52, 47, 45, 39, 47, 44.47, 28, 36, 39, 27.67, 43, 37, 41, 52, 50, 37, 30, 30, 27.4, 33, 40, 35.51, 38, 35.43, 38, 37, 42, 28.49, 27, 47, 32.45, 30, 42, 36.39, 29.62, 27, 29.55, 32, 46, 44, 29.42, 26.72, 32, 44, 34.61, 37, 45, 45, 38.44, 42, 50, 34, 47.5, 51, 30, 31.56, 34, 42, 45, 39, 30, 42, 34, 50, 27.33, 38, 41.55, 32.43, 40, 40, 50, 39, 33.53, 26, 25.73, . . . , 27, 30, 22.59, 38, 33, 26.58, 23.48, 32.62, 40, 20.33, 27.4, 33.44, 35.42, 33.34, 31, 38, 34, 21, 21, 38, 36].

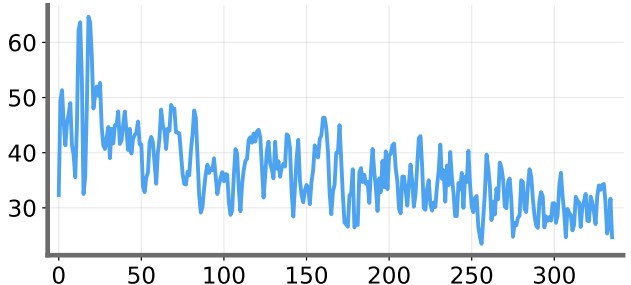

**Options:**
*A.* ATM daily cash withdrawals over a year with an annual festival (365 daily samples).
*B.* Weekend music festival causing a surge in noise levels (96 hourly samples).
*C.* Hourly restaurant food orders during a 2-week local food festival (336 hourly samples).
*D.* NYC daily taxi pick-ups over a year with a subway strike (365 daily samples).

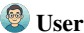 **TIMEOMNI-1**

```
<think>
```
**1) Length check.**
Observed length is 336. Options expect: A= 365, B= 96, C= 336, D= 365. *Only C matches exactly.*
**2) Magnitude sanity.**
Values span ≈ 15–92, reasonable for *hourly* food orders in a busy restaurant (C). NYC daily taxi rides (D) are typically in the thousands; A and B already fail length.
**3) Shape/pattern.**
Fluctuating baseline with intermittent surges (e.g., 92) and no single prolonged spike—consistent with repeated demand bursts during a local festival (C).
**4) Event alignment.**
A 2-week festival plausibly produces multiple hourly peaks across days (C), rather than one annual or weekend-only spike (A/B), and not daily-year scale (D).
**5) Double-check.**
2 weeks × 24 hours = 336 ⇒ consistent with (C).
```
</think>
<answer>C</answer>
```

Table 14: Task2 (Causality Discovery) example with options and reasoning process.

**Task2: Causal Discovery**

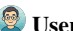 **User**
You are given two time series related to river discharge measurements, expressed in $m^3/s$. Through causal discovery methods, we aim to identify potential causal relationships between different measuring stations from time-series data alone. The time series of A463 is: [4.12, 3.8, 3.44, 3.11, 2.97, 2.85, 2.79, 2.76, 2.63, 2.74, 2.69, 2.77, 2.75, 2.79, 2.97, 3, 3.09, 3.18, 3.24, 3.16, 3.16, 3.37, 3.37, 3.54, 3.56, 3.51, 3.43, 3.37, 3.37, 3.21, 3.12, 3.07, 3.16, 3.12, 3.16, 3.15, 2.95, 3, 2.95, 2.84, 2.75, 2.74, 2.74, 2.74, 2.74, 2.74, 2.6, 2.56, 2.56, 2.53, 2.54, 2.52, 2.33, 2.33, 2.33, 2.4, 2.61, 2.9, 3.22, 3.45, 3.58, 3.95], the time series of 4PRY is [7.55, 7.31, 7, 6.78, 6.5, 6.2, 6.08, 6, 6, 5.91, 5.82, 5.78, 5.84, 5.92, 5.88, 5.89, 5.82, 6, 6.25, 6, 5.94, 5.9, 5.9, 5.89, 5.89, 6, 6, 5.9, 5.74, 5.81, 5.67, 5.55, 5.35, 5.29, 5.14, 5.11, 5.18, 5.12, 5.11, 5.07, 4.89, 4.81, 4.82, 4.7, 4.69, 4.61, 4.67, 4.58, 4.45, 4.55, 4.42, 4.44, 4.32, 4.28, 4.37, 4.53, 4.5, 4.62, 4.84, 5, 5.43, 5.74]. Please identify the causal relationships between the two measurement stations? The data is collected every 12 hours from 2020-01-01 to 2020-01-31 totally 62 points each series.

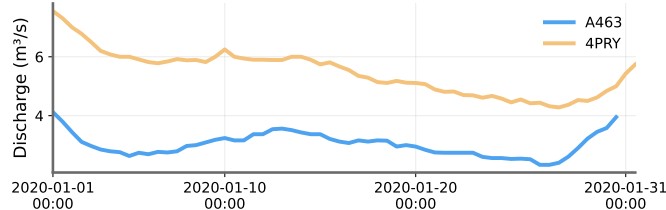

**Options:**
*A.* A463 is the cause and 4PRY is the effect
*B.* A463 and 4PRY are not causal.
*C.* 4PRY is the cause and A463 is the effect

**TIMEOMNI-1**
```
<think>
```
**1) Trend consistency (perceptual matching).**
Both A463 and 4PRY start with a gradual decrease over the first 10–12 points.
Both show a rise around point 15–20, peaking at similar positions (A463 rises from 2.7 to 3.5, 4PRY from 5.8 to 6).
Both then decline after their peaks, with a secondary smaller rise around point 45–50 (A463: 2.5 to 3.9; 4PRY: 4.3 to 5.7).
The overall shapes match: both series are "decline → flat → rise → fall → rise" with similar timing.
No major segment where one rises while the other falls or stays flat.
**2) Key fluctuation alignment (peak synchrony).**
Major peaks occur nearly simultaneously (points ∼19–20: A463 ≈ 3.37; 4PRY ≈ 5.9).
Secondary peaks/dips are aligned within 1–2 steps.
No meaningful lag ($>2$ steps) between corresponding changes.
**3) Direction of causality (hydrologic magnitude heuristic).**
Mean level: $\mathrm{mean}(A463) < \mathrm{mean}(4PRY)$ (A463 mostly 2.5–3.5; 4PRY mostly 4.5–6.5).
Hydrology rule-of-thumb: smaller tributaries feed larger channels $\Rightarrow$ upstream (smaller) $\rightarrow$ downstream (larger).
```
</think>
<answer>A</answer>
```

Table 15: Task3 (Event-aware Forecasting) example with reasoning process.

---

**Task3: Event-aware forecasting**

---

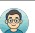 **User**

Over the past 48 hours (from `2015-05-29 00:00:00` to `2015-05-30 23:00:00`), hourly taxi pickups were:
[26, 22, 9, 6, 1, 6, 3, 8, 11, 18, 11, 28, 22, 22, 26, 33, 44, 49, 76, 91, 85, 53, 69, 133, 116, 64, 28, 7, 8, 7, 1, 3, 12, 10, 22, 22, 35, 51, 37, 59, 57, 61, 50, 87, 54, 72, 56, 49].
Significant event in this window:
   `2015-05-29 17:00`   BROOKLYN BOXING: Amir Khan vs. Chris Algieri & Fortuna vs. Vasquez.
In the next 24 hours: *2015-05-31 19:30 Ed Sheeran*.

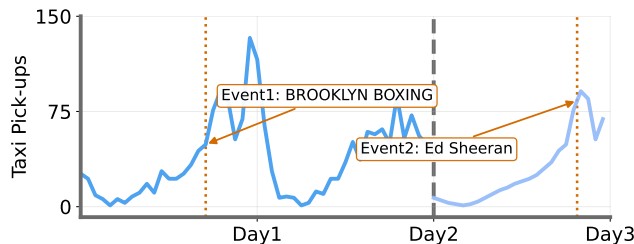

---

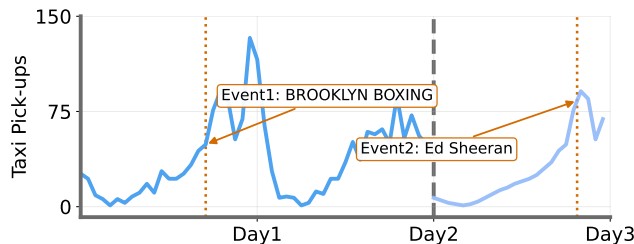 **TIMEOMNI-1**

`<think>`
**1) Identify Baseline Patterns and Trends.**
The data covers two days (48 hourly values). There is a clear daily pattern: low pickups overnight (midnight–6am), rising through morning, peaking late afternoon/evening. On both days, there is a pronounced spike around 17:00–20:00, likely due to the boxing event. After the spike, pickups drop sharply, then gradually decrease into the night.
**2) Estimate Event Impact.**
The `2015-05-29 17:00` boxing event caused a large surge: 17: 44, 18: 49, 19: 76, 20: 91, 21: 85, 22: 53, 23: 69, then post-event dispersal 0–3h: 133, 116, 64, 28; well above baseline (~20–30/h).
**3) Context for Next Day.**
Next event: `2015-05-31 19:30` Ed Sheeran concert. Expect similar pre-event buildup, a sharp spike during/after, then decline. No other events.
**4) Forecast Sequence.**
Use the prior non-event pattern as baseline and overlay an event-shaped spike at the concert time: Overnight (0–6) low (1–8); Morning (7–11) rising (8–18); Early afternoon (12–16) moderate (18–33); Pre-event (17–18) increase (33–44); Event spike (19–21) large (76–91); Post-event (22–23) decline (53–69).
**5) Finalize Sequence.**
For each hour, use baseline + event effect if applicable.
`</think>`
`<answer>`
$\big[$7, 5, 3, 2, 1, 2, 4, 7, 10, 13, 15, 18, 20, 22, 25, 30, 35, 44, 49, 76, 91, 85, 53, 69$\big]$
`</answer>`

Table 16: Task4 (Decision Making) example with reasoning process.

**Task4: Decision-making**

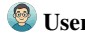 **User**

You manage a home energy storage system with battery capacity $18\,\text{kWh}$, current state-of-charge (SoC) $5\,\text{kWh}$, max charging power $5\,\text{kW}$, max discharging power $10\,\text{kW}$. Historical 48-hour hourly load (kWh) from 2023-07-10 00:00 to 2023-07-11 23:00 is given. Tomorrow's 24-hour pricing: off-peak $0.22/kWh (hours 0–14, 20–23), peak $0.54/kWh (hours 15–19). Based on historical usage, predict tomorrow's load and select the optimal 24-hour battery strategy from Options A–D (hour indices 0–23).

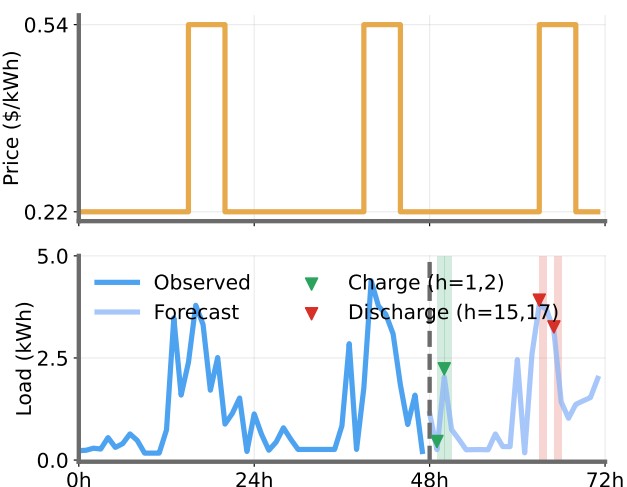

**Options:**
**A.** Charge: { 5, 11 };   Discharge: { 2, 18 }
**B.** Charge: { 1, 2 };   Discharge: { 15, 17 }
**C.** Charge: { 2, 10 };   Discharge: { 18, 19 }
**D.** Charge: { 11, 14 };   Discharge: { 16, 18 }

**TIMEOMNI-1**

```
<think>
```
**1) Load forecast.**
Mirror the most recent day (hours 24–47) as tomorrow's profile: $[1.42, 3.16, 2.06, 1.03, 1.81, 1.02, 0.71, 0.34, 0.62, 0.75, 0.87, 2.44, 3.71, 1.98, 4.86, 1.37, 1.00, 0.65, 2.40, 0.40, 2.32, 2.51, 1.59, 0.76]$.
**2) Prices & principle.**
Off-peak: hours 0–14, 20–23 ($0.22/kWh). Peak: hours 15–19 ($0.54/kWh). Charge in off-peak, discharge in peak to maximize spread $\Delta p = 0.54 - 0.22 = 0.32\,\text{\$/kWh}$.
**3) Feasibility & SoC constraints.**
Capacity $18\,\text{kWh}$; initial SoC $5\,\text{kWh}$; max charge $5\,\text{kWh/h}$; max discharge $10\,\text{kWh/h}$.
- $A$: Discharge at hour 2 (off-peak) $\Rightarrow$ violates principle $\Rightarrow$ exclude.
- $B$: Charge $\{1,2\}$ (off-peak) $\Rightarrow$ SoC: $5\to10\to15$; Discharge $\{15,17\}$ (peak).
- $C$: Charge $\{2,10\}$ $\Rightarrow$ SoC: $5\to10\to15$; Discharge $\{18,19\}$ (peak).
- $D$: Charge $\{11,14\}$ $\Rightarrow$ SoC: $5\to10\to15$; Discharge $\{16,18\}$ (peak).
**4) Discharge amounts & savings (bounded by load and $10\,\text{kWh/h}$).**
Peak-hour loads (forecast):
h15= 4.86, h16= 1.37, h17= 1.00, h18= 0.65, h19= 2.40.
- $B$: discharge at $\{15,17\}$ $\Rightarrow$ $4.86+1.00 = 5.86\,\text{kWh}$;  saving $= 5.86 \times 0.32 = \$1.8752$. (*If* discharging also at h16 is allowed as listed, then $4.86 + 1.37 + 1.00 = 7.23\,\text{kWh}$ $\Rightarrow$ \$2.3136.)
- $C$: $\{18,19\}$ $\Rightarrow$ $0.65 + 2.40 = 3.05\,\text{kWh}$ $\Rightarrow$ \$0.976.
- $D$: $\{16,18\}$ $\Rightarrow$ $1.37 + 0.65 = 2.02\,\text{kWh}$ $\Rightarrow$ \$0.6464.
All satisfy SoC bounds (start $5\,\text{kWh}$, charge to $15\,\text{kWh}$, discharge $\leq$ available).
```
</think>
```
```
<answer>B</answer>
```

