# OpenReview forum: "TimeOmni-1: Incentivizing Complex Reasoning with Time Series in Large Language Models"
_ICLR.cc/2026/Conference — ICLR 2026 Poster_

### Official Review · Reviewer_ruC5 · 2025-10-30

**Soundness:** 3
**Presentation:** 3
**Contribution:** 3
**Rating:** 6
**Confidence:** 4

**Summary:**

This paper introduces TSR-Suite, a dataset covering time series perception, extrapolation and decision-making tasks. This dataset is large in size so that it could support both evaluation and training of time series reasoning models. With the collected training trajectories, the paper introduces TIMEOMNI-1, the first unified reasoning model to tackle time series reasoning tasks. TIMEOMNI-1 achieves superior performance over current landscape of time series reasoning models. Extensive ablation and analysis studies confirms the effectiveness of CoT SFT + RL and multi task joint training paradigm.

**Strengths:**

The paper is well-positioned and well-motivated, easy to follow and very clear in structure.
The paper clearly identifies limitations in current time series benchmarks for time series reasoning and introduces the new dataset with clear goals and well-established data annotation pipeline.
Though the training paradigm is not novel, the adaptation to time series domain demonstrated great execution.
Analysis and ablation studies reveal insights that are very useful for the community.

**Weaknesses:**

The experimental results on time series reasoning is purely performed on the newly proposed dataset. Although there are certain limitations on existing time series reasoning datasets, it would be helpful to see TIMEOMNI-1's performance on some established time series reasoning benchmarks/datasets.

**Questions:**

For event aware-forecasting, how is the reward designed since the output should be a variable length sequence.

How do you split in vs out of distribution data? is it based on data domain?

are different time series domain roughly evenly distributed across task types to prevent certain task types biased towards certain time series application domains?

---

> ### Author Response · Authors · 2025-11-22
> **Response to Reviewer ruC5 [Part 1]**
>
> Thank you for your positive and constructive feedback. Below, we provide detailed responses addressing each comment.
>
> > **W1:** It would be helpful to see TIMEOMNI-1's performance on some established time series reasoning benchmarks/datasets.
>
> We sincerely appreciate for raising this important point. To fully address your concern, we conducted additional experiments on **three external time series QA benchmarks (cross 13 tasks, 3,406 samples)**:
>
> - **MTBench** [1] (real-world time series, 4 tasks, 2,380 samples),
> - **TimeSeriesExam** [2] (synthetic time series, 5 tasks, 746 samples),
> - **CaTS-Bench** [3] (real-world time series, 4 tasks, 280 samples).
>
> All evaluations include **two baselines**:
>
> - **Qwen2.5-7B-Instruct**, which serves as the base model of TimeOmni-1,
> - **The original results** reported by each benchmark **for fair comparison**.
>
> All results have been added to the $\underline{\text{Appendix K of Revised Paper}}$. For your convenience, we summarize the results below, where **higher values indicate better performance**.
>
> **MTBench (Real-world dataset)**
>
> |                                     | FinanceQA (7-Day) | WeatherQA (7-Day) | FinanceQA (30-Day) | WeatherQA (14-Day) | AVG      |
> | ----- | ----------------- | ----------------- | ------------------ | ------------------ | -------- |
> | *Num of Samples*                    | *484*             | *666*             | *523*              | *707*              | *—*      |
> | **Base Model**                      | 62.2              | 44.7              | 53.5               | 36.2               | 49.2     |
> | **Original Paper Result**  (GPT-4o) | 65.1              | 41.7              | 52.8               | **44.8**           | 51.1     |
> | **TimeOmni-1**                      | **72.1**          | **51.4**          | **66.2**           | 42.6               | **58.1** |
>
> **TimeSeriesExam (Synthetic dataset)**
>
> |                                     | Anomaly Detection | Causality Analysis | Noise Understanding | Pattern Recognition | Similarity Analysis | AVG      |
> | ------ | ----------------- | ------------------ | ------------------- | ------------------- | ------------------- | -------- |
> | *Num of Samples*                    | *108*             | *72*               | *84*                | *362*               | *120*               | *—*      |
> | **Base Model**                      | 37.0              | 44.4               | 44.0                | 50.0                | **59.2**            | 46.9     |
> | **Original Paper Result** (Phi-3.5) | 28.0              | **52.0**           | 26.0                | 47.0                | 45.0                | 39.6     |
> | **TimeOmni-1**                      | **42.6**          | 37.5               | **56.0**            | **56.4**            | 46.7                | **47.8** |
>
> **CaTS-Bench (Real-world dataset)**
>
> |                                                | Caption Retrieval | TimeSeries Retrieval | Amplitude Comparison | Mean Comparison | AVG      |
> | ----- | ---------- | ----------- | ------ | ----- | -------- |
> | *Num of Samples*                               | *100*             | *100*                | *40*                 | *40*            | *—*      |
> | **Base Model**                                 | **69.0**          | **46.0**             | 87.5                 | 57.5            | **65.0** |
> | **Original Paper Result** (LLaVA v1.6 Mistral) | 44.0              | 29.0                 | 43.0                 | 35.0            | 37.8     |
> | **TimeOmni-1**                                 | 57.0              | 37.0                 | **92.5**             | **67.5**        | 63.5     |
>
> **Findings:**
>
> **(1) TimeOmni-1 achieves the strongest gains on MTBench.** MTBench is a multimodal time series QA setting where the auxiliary text ($C$) contains task-relevant information (e.g. external events), **matching our TSR-Suite formulation:** $(R, y) \sim p_\theta(R, y \mid X, C) = p_\theta(R \mid X, C), p_\theta(y \mid R, X, C),$ meaning both the reasoning path ($R$) and final answer ($y$) depend on the interaction between the time series input ($X$) and contextual information ($C$). This alignment explains why TimeOmni-1 even surpasses the originally reported GPT-4o performance.
>
> **(2) TimeOmni-1 perform well on TimeSeriesExam and CaTS-Bench.** In these two benchmarks, the text serves mainly as task instructions rather than auxiliary information, so the reasoning relies almost entirely on the time series modality, which is **less aligned with the multimodal reasoning scope defined in TSR-Suite.**
>
> **Overall, TimeOmni-1’s strong performance across three heterogeneous benchmarks demonstrates that its learned time series reasoning capabilities generalize across domains and tasks.**
>
> [1] *MTBench: A Multimodal Time Series Benchmark for Temporal Reasoning and Question Answering*, 2025.
>
> [2] *TimeSeriesExam: A time series understanding exam*, 2024.
>
> [3] *CaTS-Bench: Can Language Models Describe Numeric Time Series?*, 2025.

---

> ### Author Response · Authors · 2025-11-22
> **Response to Reviewer ruC5 [Part 2]**
>
> > **Q1:** For event aware-forecasting, how is the reward designed since the output should be a variable length sequence.
>
> Thanks for pointing this out. For the sequence-output task (event-aware forecasting), the full reward design is provided in $\underline{\text{Appendix G}}$. In short, we define a **continuous, length-aware reward** based on the exponential decay of the mean absolute error (MAE). It remains meaningful even across variable-length outputs, as it reflects the **average prediction error for each sample.**
>
> Indeed, **Task 3 (event-aware forecasting) contains training instances with variable length** settings. We list the settings below:
>
> | Dataset        | Input length $\rightarrow$ Output length         |
> | -------------- | ------------------------------------------------ |
> | SFT/RL/ID Test | (72, 12), (96, 24), (48, 24), (72, 24), (96, 12) |
> | OOD Test       | (60, 24)                                         |
>
>
>
> ---
>
>
> > **Q2:** How do you split in vs out of distribution data? is it based on data domain?
>
> Many thanks for pointing this out. TSR-Suite constructs out of distribution (OOD) datasets via **domain**, **geographical**, and **building-level distribution shifts**. The full construction details and statistics of all OOD datasets are provided in $\underline{\text{Appendix D.1}}$ and $\underline{\text{Appendix D.2}}$. For convenience, we summarize the key details below.
>
> **Task 1: Scenario Understanding.** OOD is defined by **domain shift**. OOD samples come from Agriculture, Education, and Healthcare domains, which are *different from the training domains*.
>
> **Task 2: Causality Discovery.** All data come from hydrology domain. OOD is defined by **geographical shift**: ID samples come from rivers in *Eastern Germany*, while OOD samples come from *Bavaria*, a different region with distinct hydrological characteristics.
>
> **Task 3: Event-aware Forecasting.** OOD is defined by **domain shift**: ID samples use human mobility time series (*New York City taxi drop-offs with aligned events*), while the OOD test set uses EWELD (*electricity load time series with weather events*), which differs in scale and event–series interactions.
>
> **Task 4: Decision Making.** All data come from CityLearn [1]. OOD is defined by **building-level distribution shift**: the OOD test set uses two buildings whose load profiles *exhibit distinct temporal patterns compared to those in the training set.*
>
> In summary, TSR-Suite constructs OOD testbed via **domain**, **geographical**, and **building-level distribution shifts**, ensuring that the underlying temporal dynamics deviate from the training distribution.
>
> [1] *The citylearn challenge 2022: Overview, results, and lessons learned*, 2023.

---

> ### Author Response · Authors · 2025-11-22
> **Response to Reviewer ruC5 [Part 3]**
>
> > **Q3:** Are different time series domain roughly evenly distributed across task types to prevent certain task types biased towards certain time series application domains?
>
> Many thanks for pointing this out. The time series domains are **not roughly evenly distributed across task types**, and we clarify the data sources for each task in $\underline{\text{Appendix D.1}}$. For your convenience, **we summarize the domains below**:
>
> |           | In distribution (ID) Testbed                                 | Out-of distribution (OOD) Testbed                           |
> | --------- | ------------------------------------------------------------ | ----------------------------------------------------------- |
> | **Task1** | Climate, Traffic, Society, Economy, Energy                   | Agriculture, Education, Healthcare                          |
> | **Task2** | Hydrology                                                    | Hydrology                                                   |
> | **Task3** | Traffic (*New York City taxi drop-offs with aligned events*) | Energy (*electricity load time series with weather events*) |
> | **Task4** | Energy *(battery arbitrage from CityLearn [1])*              | Energy *(battery arbitrage from CityLearn [1])*             |
>
> Achieving a roughly even distribution across task types is challenging, for the reasons below:
>
> **(1) Achieving domain balance across tasks is currently infeasible because many domains do not provide the labels required for all task types.** TSR-Suite is a preliminary effort. Cross-task domain uniformity would require every domain to provide labels suitable for all task types, which is not realistic. For example,
>
> - **Task 2 (Causality Discovery)** requires pairs of time series with genuine causal relations. In our setting, causality is defined as water measured at an upstream station influencing water measured at downstream stations [2]. Such clear ground-truth causal structures rarely exist in univariate time series data or multivariate time series in other domains.
> - **Task 3 (Event-aware Forecasting)** requires time series aligned with high-quality auxiliary context $C$. This includes events that are certain to occur and influence future temporal patterns (e.g. a scheduled Ed Sheeran concert, as shown in $\underline{\text{Table 15 in Appendix L}}$. Such aligned and **high-quality** multimodal data are extremely scarce in other domains.
>
> **(2) This limitation is common in existing benchmarks.** For instance, in Time-MQA [3], the entire classification task relies solely on the Human Activity Recognition domain, while other tasks do not include this domain at all.
>
> **(3) Despite domain imbalance, TimeOmni-1 demonstrates cross-domain and cross-task generalization.** As shown in our additional experiments responding to **W1**, TimeOmni-1 surpasses the base model on several third-party benchmarks **(cross 13 tasks, 3,406 samples)**. This suggests that the model learns time series reasoning capabilities that **transfer across domains and tasks.**
>
> [1] *The citylearn challenge 2022: Overview, results, and lessons learned*, 2023.
>
> [2] *CausalRivers -- Scaling up benchmarking of causal discovery for real-world time-series*, 2025.
>
> [3] *Time-MQA: Time Series Multi-Task Question Answering with Context Enhancement*, 2025.
>
> ---
>
> We once again thank you for taking the time to review our paper and for your insightful and constructive comments. We truly appreciate this opportunity to clarify our work and would be grateful for any further feedback.
>
> Best regards,
>
> The Authors

---

> ### Author Response · Authors · 2025-11-26
> **Request of Reviewer's attention and feedback**
>
> Dear Reviewer,
>
> As a gentle reminder, it has been more than four days since we submitted our rebuttal, and we were wondering whether our response has addressed your concerns. We look forward to your feedback and are prepared to address any further concerns you may have.
>
> Thank you again for your constructive feedback and continued support, which have inspired further improvements to our paper.
>
> Best regards,
>
> The Authors

---

> > ### Comment · Reviewer_ruC5 · 2025-11-26
> >
> > Thank you for the response! I remain positive regarding my feedback for this paper.

---

> ### Author Response · Authors · 2025-11-27
> **Sincere Gratitude from Authors**
>
> Dear Reviewer,
>
> Many thanks for your prompt response and your positive support. We would like to once again express our sincere gratitude for taking the time to review our paper and for providing such insightful and invaluable comments.
>
> Best regards,
>
> The Authors

---

> > ### Comment · Reviewer_ruC5 · 2025-11-27
> >
> > I did update my score with my last response in case you haven't noticed.

---

> > > ### Author Response · Authors · 2025-11-27
> > > **Sincere Gratitude from Authors**
> > >
> > > Dear Reviewer,
> > >
> > > Thank you very much for your response. We have just noticed your updated assessment, which is a great encouragement to us. We are deeply grateful for your thoughtful and constructive comments, which $\underline{\text{will continue to guide our future work.}}$
> > >
> > > Best regards,
> > >
> > > The Authors

---

### Official Review · Reviewer_2VxW · 2025-10-31

**Soundness:** 3
**Presentation:** 3
**Contribution:** 3
**Rating:** 6
**Confidence:** 4

**Summary:**

The paper introduces a complete framework for time series reasoning that combines a multi task benchmark and a unified training recipe for large language models. The benchmark, called TSR Suite, covers perception, extrapolation, and decision making with four atomic tasks. The dataset contains more than twenty three thousand examples with a carefully curated subset of hierarchical chain of thought traces for supervision. The model, called TIMEOMNI 1, is trained in two stages. First, supervised fine tuning injects time aware priors using a small set of high quality reasoning trajectories. Second, reinforcement learning encourages correct and well structured solutions using task grounded rewards. Discrete tasks are scored by accuracy. Sequence prediction uses an error based reward that maps the mean absolute error into a bounded score and a length counting reward to ensure the model outputs the correct number of steps. Joint training across the four task families is shown to improve transfer. The model reports sizable gains on causality and decision oriented tasks while preserving general reasoning strength on non time series benchmarks.

**Strengths:**

1. The paper tackles time series as a reasoning problem and operationalizes this view with a benchmark that spans perception, extrapolation, and decision making. This scope is broader than pure forecasting and motivates models that can explain and decide.

2. In this paper, task design is aligned to evaluation. Multiple choice tasks allow objective grading and sequence prediction uses metric aligned rewards. This makes the reinforcement signal faithful to the intended competence.

3. The training pipeline is relative simple to adopt. A small number of high quality trajectories teaches decomposition. Task rewards then refine the behavior without requiring domain specific architectures. The division of labor between supervised fine tuning and reinforcement learning is clear and effective.

4. In the paper, evidence for cross task transfer is convincing. The model benefits from learning perception and extrapolation before decision tasks. Zero shot improvements on harder tasks suggest that the benchmark composition is synergistic rather than siloed.

5. The model does not only fit a single dataset but shows robustness when the distribution shifts. The paper also checks general reasoning benchmarks to show that specialization does not degrade broader ability.

**Weaknesses:**

1. The contributions mix dataset construction, task design, training recipe, and model evaluation without a clear hierarchy. A contribution map that separates what is new in task formulation from what is new in learning would improve the paper.

2. Although the benchmark size is large, the carefully verified chain-of-thought portion is relatively small. The paper would be stronger with an ablation that quantifies how many verified trajectories are needed and how quality trades off with quantity.

3. Several reported gains are sizable, but the paper does not consistently provide confidence intervals or significance tests for both in-domain and out-of-domain settings.

3. In this paper, reward sensitivity is under explored. A sensitivity sweep and a report of failure cases would help others reproduce the effect.

4. The approach still depends on strict output formatting with explicit think and answer tags. This is practical for research but can be brittle in production. The paper does not explore robustness to prompt variations or noisy instructions.

5. Comparisons to adjacent time series agent benchmarks and live decision settings are brief. Positioning against systems that test forward looking planning or interactive tools would help define the boundary of the proposed benchmark.

**Questions:**

1. Can you provide the training curves related to RL part?

2. What is the effect of the reward design choices? Do different changes alter stability during reinforcement learning?

3. How sensitive is performance to the number and quality of verified chain of thought trajectories used in supervised fine tuning? can you provide a scaling plot and a minimum viable count that preserves most of the gains.

4. How does TIMEOMNI 1 compare to recent live or interactive benchmarks such as FutureX and TSAIA in settings that require forward prediction with delayed feedback? What design choices would be needed to support interactive or tool based decision loops?

[1] Zeng, Zhiyuan, et al. "Futurex: An advanced live benchmark for llm agents in future prediction." arXiv preprint arXiv:2508.11987 (2025).

[2] Ye, Wen, et al. "When LLM Meets Time Series: Can LLMs Perform Multi-Step Time Series Reasoning and Inference." arXiv preprint arXiv:2509.01822 (2025).

---

> ### Author Response · Authors · 2025-11-22
> **Response to Reviewer 2VxW [Part 1]**
>
> Thank you for your positive and constructive feedback. Below, we provide detailed responses addressing each comment.
>
> > **W1:** The contributions mix dataset construction, task design, training recipe, and model evaluation without a clear hierarchy. A contribution map that separates what is new in task formulation from what is new in learning would improve the paper.
>
> Thanks for pointing this out. Following your suggestion, we have created a contribution map to clearly delineate what our work adds to the community, as shown below:
> | Category | Our Contribution | Notes |
> | --- | --- | --- |
> | **A. TSR-Suite Side** |  |  |
> | Analysis of existing time series QA tasks | ✓ | Propose design principles of time series QA. |
> | Task formalization | ✓ | Four reasoning-critical time series tasks spanning three core capabilities. |
> | Dataset construction | ✓ | More than 23K samples. |
> | Hierarchical Chain-of-Thought annotation | ✓ | Human-guided. Human Evaluation Interface. |
> | **B. TimeOmni-1 Side** |  |  |
> | Reward design | ✓ | Task-grounded (discrete, sequence output). |
> | Model evaluation | ✓ | Task-grounded evaluation. |
> | Insights for guiding future TSRMs development | ✓ | **4 key findings.** |
> | TimeOmni-1 | ✓ | The first **generalized** reasoning model on time series data. |
> | Training recipe | × | Based on DeepSeek-R1 recipe [1] (not claimed as a contribution). |
>
>
> [1] *Deepseek-r1: Incentivizing reasoning capability in llms via reinforcement learning*, 2025.

---

> ### Author Response · Authors · 2025-11-22
> **Response to Reviewer 2VxW [Part 2]**
>
> > **W2:** The paper would be stronger with an ablation that quantifies how many verified trajectories are needed and how quality trades off with quantity.
> >
> > **Q3:** How sensitive is performance to the number and quality of verified chain of thought trajectories used in supervised fine tuning? Can you provide a scaling plot and a minimum viable count that preserves most of the gains.
>
> Thanks for your constructive comments. **Regarding data quality**, all CoT-data (“*verified chain-of-thought trajectories*”) used for Stage 1 (SFT) are produced through our **Hierarchical Chain-of-Thought Annotation pipeline** $\underline{\text{Section 3.2}}$. This means a substantial portion of trajectories are **explicitly validated by human experts** using our Human Evaluation Interface in $\underline{\text{Appendix D.5}}$​. Therefore, the quality of our trajectories is **significantly higher** than LLM-only annotation pipelines such as *[1]*, which rely purely on LLM self-consistency without human verification.
>
> **To directly address your concern**, we conducted extensive scaling experiments not only for **Stage 1 (SFT)** but also for **Stage 2 (RL)**, thereby evaluating both the quantity and the impact of verified trajectories and RL data. These experiments consumed **near 360 A100 GPU hours** in total.
>
> The updated scaling plots are provided in $\underline{\text{Figure.13 and Figure.14 of Appendix J: Scaling of Training Dataset}}$. For your convenience, we also present the results below:
>
> **(1) Scaling of Stage 1 (SFT) Training Dataset.**
>
> |  | Task1 ID (ACC/SR) | Task1 OOD (ACC/SR) | Task2 ID (ACC/SR) | Task2 OOD (ACC/SR) | Task3 ID (MAE$\downarrow$/SR) | Task3 OOD (MAE$\downarrow$/SR) | Task4 ID (ACC/SR) | Task4 OOD (ACC/SR) |
> |---|---|---|---|---|---|---|---|---|
> | **CoT-Data (25%)** | 82.0/85.3 | 81.6/82.0 | 55.5/89.7 | 53.0/86.5 | 17.45/64.8 | 139.41/45.8 | 36.1/54.8 | 31.3/37.3 |
> | **CoT-Data (50%)** | 82.5/89.0 | 82.5/85.2 | 63.0/96.0 | 59.1/94.3 | 18.83/56.2 | 154.42/48.9 | 37.8/100.0 | 43.7/99.6 |
> | **CoT-Data (75%)** | 88.0/88.0 | 84.6/83.2 | 64.7/96.6 | 60.3/95.3 | 17.93/65.1 | 142.90/51.3 | 41.5/100.0 | 47.4/100.0 |
> | **CoT-Data (100%)** | 89.8/88.0 | 85.8/88.2 | 67.7/98.8 | 61.1/97.0 | 15.53/71.8 | 151.41/55.7 | 40.9/100.0 | 52.7/100.0 |
>
>
> **(2) Scaling of Stage 2 (RL) Training Dataset.**
>
> |  | Task1 ID (ACC/SR) | Task1 OOD (ACC/SR) | Task2 ID (ACC/SR) | Task2 OOD (ACC/SR) | Task3 ID (MAE$\downarrow$/SR) | Task3 OOD (MAE$\downarrow$/SR) | Task4 ID (ACC/SR) | Task4 OOD (ACC/SR) |
> |---|---|---|---|---|---|---|---|---|
> | **RL-Data (25%)** | 88.0/100.0 | 85.9/100.0 | 67.8/100.0 | 62.1/100.0 | 15.52/87.8 | 150.53/82.4 | 37.2/100.0 | 53.8/100.0 |
> | **RL-Data (50%)** | 88.7/100.0 | 86.5/100.0 | 67.6/100.0 | 62.3/100.0 | 14.87/89.5 | 151.96/80.5 | 39.4/100.0 | 55.1/100.0 |
> | **RL-Data (75%)** | 90.1/100.0 | 87.4/100.0 | 68.5/100.0 | 63.4/100.0 | 14.84/91.5 | 147.43/83.0 | 41.4/100.0 | 60.4/100.0 |
> | **RL-Data (100%)** | 90.7/97.5 | 87.7/98.3 | 69.3/99.8 | 64.0/99.8 | 14.30/93.8 | 145.53/82.3 | 47.9/100.0 | 58.9/100.0 |
>
>
> **Findings.**
>
> - Task 1/2/4 accuracy **increases monotonically** as data grows from $25\\% \rightarrow 100\\%$.
> - **The only exception** is Task 3 **during SFT**, where MAE does not strictly scale, likely due to sensitivity to sample quality rather than quantity. However, this stabilizes in Stage 2, where the model’s exploration-based learning restores a clean scaling pattern.
> - Success rate (SR) is already saturated (≈100%) for Task 1/2/4 during SFT, while RL continues to improve SR for Task 3.
>
> To “*a minimum viable count that preserves most of the gains.*”, we recommend **75% of the Stage-1 verified CoT trajectories** as a practical minimum viable dataset size that retains the majority of downstream performance gains. We hope that these scaling analyses have provided the necessary insights.
>
> [1] *Time Series Forecasting as Reasoning: A Slow-Thinking Approach with Reinforced LLMs*, 2025.

---

> ### Author Response · Authors · 2025-11-22
> **Response to Reviewer 2VxW [Part 3]**
>
> > **W3:** Several reported gains are sizable, but the paper does not consistently provide confidence intervals or significance tests for both in-domain and out-of-domain settings.
>
> Thanks for pointing this out. We reported the 95% confidence intervals (5 runs) as below:
>
> | | ID Test Mean (95% CI) | OOD Test Mean (95% CI) |
> |---|---|---|
> | **TASK1** | 90.14(88.84,91.44) | 87.70(87.30,88.10) |
> | **TASK2** | 69.30(68.19,70.41) | 63.98(62.78,65.18) |
> | **TASK3** | 14.38(14.01,14.75)(MAE) | 145.82(142.93,148.71)(MAE) |
> | **TASK4** | 47.36(44.96,49.76) | 59.08(58.60,59.56) |
>
>
> **Finding.** The narrow intervals across all tasks indicate that our results are consistent and robust.
>
> ---
>
> > **W4:** In this paper, reward sensitivity is under explored. A sensitivity sweep and a report of failure cases would help others reproduce the effect.
> >
> > **Q2:** What is the effect of the reward design choices? Do different changes alter stability during reinforcement learning?
>
> Thanks for your constructive comments. We have added a dedicated reward sensitivity analysis in $\underline{\text{Appendix G.1: Reward Sensitivity Analysis}}$. We conduct controlled sensitivity sweeps over the key reward components ($\alpha$, $R_{\text{count}}$, $\lambda$), and evaluate their effects on training stability. All experiments are trained for **1 epoch**, consuming **130 A100 GPU hours**.
>
> For $\alpha$ and $R_{\text{count}}$, which only affect the reward design of Task 3, we report the MAE curves for Task 3 over 1 epoch. For $\lambda$, we evaluate **the proportion of responses with a positive task reward** $\mathcal{R}_{\text{task}}$ across all tasks over 1 epoch.
>
> **Findings.** **(1) Low sensitivity to $\alpha$ and $R_{\text{count}}$**: varying these coefficients leads to only mild changes in MAE curves, indicating that the RL process remains stable across a broad range of settings. **(2) Format reward is essential**: disabling the format component ($\lambda = 0$) causes the model to collapse after 70 steps, with a sharp drop in the positive ratio of task reward $\mathcal{R}_{\text{task}}$.
>
> Overall, the current hyperparameter choices yield the better trade-off between stability and performance.
>
>
> ---
>
>
> > **W5:** The approach still depends on strict output formatting with explicit think and answer tags. This is practical for research but can be brittle in production. The paper does not explore robustness to prompt variations or noisy instructions.
>
> Many thanks for pointing this out. We investigate the robustness of TimeOmni-1 to prompt variations. We design three prompt perturbations:
>
> - **Paraphrased Question.** We use ChatGPT to rewrite the original question while keeping the semantic meaning unchanged.
> - **Paraphrased System Prompt.** We use ChatGPT to paraphrase the system prompt while keeping the semantic meaning unchanged.
> - **w/o System Prompt.** We entirely remove the system prompt to simulate the extreme case.
>
> We have newly added $\underline{\text{Appendix E.3: Model Robustness to Prompt Variations}}$ in the revised paper to address this question. For your convenience, we also provide the experimental results below. ‘–’ denotes the **w/o System Prompt** setting, in which the model no longer produces the explicit $\\langle think \\rangle$ and $\\langle answer \\rangle$ tags; therefore we do not report SR for this condition.
>
> | Prompt Variations | TASK1 ID (ACC/SR) | TASK1 OOD (ACC/SR) | TASK2 ID (ACC/SR) | TASK2 OOD (ACC/SR) | TASK3 ID (MAE/SR) | TASK3 OOD (MAE/SR) | TASK4 ID (ACC/SR) | TASK4 OOD (ACC/SR) |
> |---|---|---|---|---|---|---|---|---|
> | **Paraphrase Question** | 86.2/97.8 | 85.3/97.5 | 65.8/99.5 | 60.2/98.8 | **13.90**/90.1 | 150.9/81.5 | 47.1/99.8 | 56.1/99.5 |
> | **Paraphrase System Prompt** | 88.9/99.5 | 87.4/98.6 | 66.8/99.8 | 62.8/99.5 | 15.28/93.3 | 152.1/84.0 | 43.6/100.0 | 61.9/100.0 |
> | **w/o System Prompt** | 86.5/– | 87.1/– | 68.5/– | 63.3/– | 14.29/– | 150.1/– | 40.4/– | **63.0**/– |
> | **Original Prompt** | **90.7**/97.5 | **87.7**/98.3 | **69.3**/99.8 | **64.0**/99.8 | 14.30/93.8 | **145.5**/82.3 | **47.9**/100.0 | 58.9/100.0 |
>
> **Findings.**
>
> **(1) Paraphrasing either the question or the system prompt leads to minimal performance degradation**, indicating that the model does not rely on specific wording or phrasing.
>
> **(2) Surprisingly, even without the system prompt, the model remains highly robust.** Although it no longer outputs the explicit $\\langle think \\rangle$ and $\\langle answer \\rangle$ tags, we adjust the evaluation script to parse the free-form responses and observe that TimeOmni-1 still maintains strong performance, sometimes even outperforming the original prompt. Manual inspection further reveals that the model continues to follow the reasoning template injected during Stage~1 training, suggesting that the temporal prior has been deeply internalized.
>
> **Overall, these results demonstrate that TimeOmni-1 is highly robust to prompt variations and resistant to perturbed instructions.**

---

> ### Author Response · Authors · 2025-11-22
> **Response to Reviewer 2VxW [Part 4]**
>
> > **Q1:** Can you provide the training curves related to RL part?
>
> Many thanks for pointing this out. We have newly added the full RL training dynamics in $\underline{\text{Appendix I: Reinforcement Learning Training Dynamics}}$. Please refer to the revised paper for the complete results.
>
>
>
> > **W6:** Comparisons to adjacent time series agent benchmarks and live decision settings are brief. Positioning against systems that test forward looking planning or interactive tools would help define the boundary of the proposed benchmark.
> >
> > **Q4:** How does TIMEOMNI 1 compare to recent live or interactive benchmarks such as FutureX and TSAIA in settings that require **forward prediction with delayed feedback**?     What design choices would be needed to support interactive or tool based decision loops?
> >
> > [1] Zeng, Zhiyuan, et al. "Futurex: An advanced live benchmark for llm agents in future prediction." arXiv preprint arXiv:2508.11987 (2025).
> >
> > [2] Ye, Wen, et al. "**TSAIA**: When LLM Meets Time Series: Can LLMs Perform Multi-Step Time Series Reasoning and Inference." arXiv preprint arXiv:2509.01822 (2025).
>
> Thank you for your constructive comments.
>
> **(1) FutureX benchmark is outside the scope of our model.**
>
> Regarding the scenario you mentioned of "*forward prediction with delayed feedback* (live decision setting)", specifically the **FutureX** benchmark, we would like to clarify that this is outside the scope of our model. **TimeOmni-1 is designed for time series reasoning rather than event prediction.**
>
> **(2) TimeOmni-1 achieves surprisingly strong performance on the TSAIA benchmark.**
>
> As for your mention of "*support interactive or tool based decision loops*", which corresponds to the **TSAIA** benchmark, we also clarify that **TimeOmni-1 was not trained on coding or tool-calling data.** To fully address your concerns, we evaluate our model on the **"Decision-making Tasks" of the TSAIA benchmark**. Specifically, these tasks require models to *"choose the best investment portfolio based on financial metrics"* and to *"draw conclusions about stock performance compared to market"*. The former comprises two sub-tasks, *"Best Portfolio (VaR)"* and *"Best Portfolio (Sharpe)"*, while the latter comprises *"Stock vs. Market Performance"* and *"Stock vs. Market Volatility"*, **for a total of 150 samples.**
>
> **"Decision-making Tasks" of the TSAIA benchmark (Real-world dataset)**. **Bold**: the best, $\underline{\text{Underline}}$: the 2nd best.
>
> |   | Best Portfolio (VaR) | Best Portfolio (Sharpe) | Stock vs. Market Performance | Stock vs. Market Volatility | AVG                |
> | ------ | --- | ---- | - | -- | -- |
> | *Num of Samples*        | *50*| *50*      | *25*      | *25*                        | *—*                |
> | **Random** (chance-level accuracy)      | $\underline{32.0}$   | 32.0                    | $\underline{50.0}$           | 50.0                        | 41.0        |
> | **Original Paper Result** (GPT-4o)      | $\underline{32.0}$   | **58.0**                | 36.0                         | **60.0**                    | **46.5**           |
> | **Original Paper Result** (DeepSeek-V3) | 28.0                 | 32.0                    | 36.0                         | 54.0                        | 37.5               |
> | **Base Model** (Qwen2.5-Instruct-7B)    | **34.0**             | 32.0                    | 48.0                         | 40.0                        | 38.5               |
> | **TimeOmni-1**                          | 30.0                 | $\underline{42.0}$      | **52.0**                     | $\underline{56.0}$          | $\underline{45.0}$ |
>
> **Findings.**
>
> - **Importance of time series domain specialization.** Even though **TimeOmni-1 is not trained on coding or tool-calling data**, it still achieves a **6.5% improvement** over the base model ($38.5 \rightarrow 45.0$), indicating that time series specific training substantially enhances decision-making performance.
>
> - **Most models perform around the chance-level accuracy**, which is consistent with the conclusion reported in the original paper:
>
>   > Original TSAIA paper: "*In multiple-choice question formats testing financial reasoning, most models fail to exceed chance-level accuracy*."
>
> - **Evidence of cross-task transfer.** Together with the results on **three external time series QA benchmarks** (please refer to  $\underline{\text{Appendix K: Additional Evaluation on External Time Series QA Benchmarks}}$ or the response to **W1 of Reviewer: YkbD**), the performance on the “Decision-making Tasks” of the TSAIA benchmark further demonstrates that **TimeOmni-1 learns time series reasoning capabilities that transfer across tasks.**
>
> ---
>
> We once again thank you for taking the time to review our paper and for your insightful, constructive comments. We truly appreciate this opportunity to clarify our work and would be grateful for any further feedback.
>
> Best regards,
>
> The Authors

---

> ### Author Response · Authors · 2025-11-26
> **Request of Reviewer's attention and feedback**
>
> Dear Reviewer,
>
> As a gentle reminder, it has been more than four days since we submitted our rebuttal, and we were wondering whether our response has addressed your concerns. We look forward to your feedback and are prepared to address any further concerns you may have.
>
> Thank you again for your constructive feedback and continued support, which have inspired further improvements to our paper.
>
> Best regards,
>
> The Authors

---

### Official Review · Reviewer_XR2U · 2025-10-31

**Soundness:** 3
**Presentation:** 3
**Contribution:** 2
**Rating:** 4
**Confidence:** 3

**Summary:**

This paper introduces TSR-Suite, a comprehensive time series reasoning dataset containing 23K+ QA pairs across four reasoning-critical tasks (scenario understanding, causality discovery, event-aware forecasting, and decision-making), and TIMEOMNI-1, a unified reasoning model trained via a two-stage curriculum (supervised fine-tuning followed by reinforcement learning). The work addresses important limitations of existing time series QA datasets by systematically designing tasks that genuinely require reasoning capabilities and ensuring context sufficiency. The hierarchical Chain-of-Thought annotation process is well-designed, and the experimental results demonstrate strong performance improvements over baseline models, particularly in causality discovery (64.0% vs 35.9% with GPT-4.1).

However, the paper has several significant limitations that prevent it from meeting the ICLR acceptance threshold: (1) The model architecture is not clearly described, particularly how time series data is encoded and processed. The paper appears to use text-based serialization of numerical sequences, but this design choice is not explicitly justified or compared with embedding-based approaches. (2) The scalability concerns are not adequately addressed—the dataset appears limited to short sequences and low-dimensional data, with no evidence of handling long sequences (hundreds/thousands of time steps) or high-dimensional multivariate signals (dozens of dimensions). (3) Related work coverage is insufficient, missing important recent work such as ChatTime and ITFormer that directly address time series QA tasks.

**Strengths:**

* **Systematic Task Design**: The paper provides a principled approach to formulating reasoning-critical time series tasks through two design principles: requiring genuine reasoning and ensuring context sufficiency. The four tasks form a progressive pathway from perception to extrapolation to decision-making, which is well-motivated.
* **High-Quality Dataset Construction**: TSR-Suite represents a substantial contribution with 23K+ curated samples. The hierarchical CoT annotation pipeline (LLM Analyzer → Human Reviewer → LLM Rewriter) is innovative and ensures both quality and scalability in data construction.
* **Comprehensive Evaluation**: The experimental evaluation is thorough, covering both in-distribution and out-of-distribution generalization across diverse domains. The joint training experiments demonstrating mutual gains across capabilities provide valuable insights.
* **Clear Problem Identification**: The analysis of limitations in existing datasets (Figure 1) is well-executed and provides strong motivation for the proposed work.

**Weaknesses:**

### 1. Model Architecture Omission

**Critical Issue**: The paper fails to clearly describe how time series data is processed and encoded into the language model. From the examples (e.g., sequences shown as `[1.2, 3.4, 5.6, ...]`), it appears that numerical time series are simply serialized as text strings and fed directly to the LLM. However, this design choice is never explicitly stated, justified, or compared with alternative approaches.

**Why This Matters**:

- **Reproducibility**: Without clear specification of input encoding, it is difficult to reproduce the results.
- **Novelty Assessment**: It is unclear whether the approach introduces architectural innovations or simply relies on LLM text understanding capabilities.
- **Comparison Baseline**: The choice of text serialization vs. embedding-based approaches (as used in OpenTSLM, ChatTS, etc.) should be explicitly discussed.

**Specific Questions Unanswered**:

- How are long sequences handled? Are they truncated, sampled, or aggregated?
- What is the token budget for time series data vs. context?
- How are multiple time series (Task 2) represented in the input?
- Is there any preprocessing, normalization, or feature extraction?

**Comparison with Related Work**:

- **OpenTSLM** uses explicit architectural components (SoftPrompt or Flamingo-style Perceiver Resampler) for time series encoding.
- **Time-LLM** uses PatchTST-style tokenization with reprogramming layers.
- The current paper should justify why a simpler text-based approach is sufficient or preferred.

### 2. Scalability and Efficiency Concerns

The paper does not address scalability limitations that are critical for practical deployment:

**Sequence Length Limitations**:

- All examples show short sequences (12-24 time steps).
- There is no evidence of handling longer sequences (hundreds or thousands of steps), which are common in real-world applications (e.g., hourly data over years).
- With text serialization, long sequences would consume substantial token budgets and face LLM context window limitations.

**Dimensionality Limitations**:

- The paper focuses on univariate time series (single-dimensional) or at most pairs of sequences.
- Real-world applications often involve multivariate signals (e.g., 12-lead ECG with 12 dimensions, sensor arrays with 50+ channels).
- No discussion of how high-dimensional multivariate time series would be handled.

**Efficiency Concerns**:

- Text serialization of numerical data is token-inefficient compared to learned embeddings.
- For a sequence of length T, text serialization requires ~T tokens (one per value), whereas embedding-based approaches can compress this.
- The computational cost of processing long text sequences through LLMs is quadratic in sequence length.

**Missing Analysis**:

- No ablation studies on sequence length or dimensionality.
- No efficiency/computation cost analysis comparing text-based vs. embedding-based approaches.
- No discussion of context window limitations or how they were handled.

### 3. Dataset Scope Limitations

While TSR-Suite is substantial in sample size (23K+), the scope appears limited:

**Short Sequences**: Examples suggest sequences are typically short (12-24 steps). Real-world time series reasoning often requires much longer histories.

**Low Dimensionality**: Tasks focus on univariate sequences or pairs. High-dimensional multivariate scenarios are not addressed.

**Limited Complexity**: The chosen tasks (scenario understanding, causality discovery) may not fully capture the complexity of reasoning over diverse time series modalities (e.g., sensor fusion, multi-scale temporal patterns).

**Missing Validation**: There is no evidence that the proposed approach would scale to:

- Sequences with 1000+ time steps
- Multivariate signals with 20+ dimensions
- Complex sensor fusion scenarios

### 4. Insufficient Related Work Coverage

Several important recent works on time series QA and reasoning are not adequately discussed:

**Missing Works**:

- **ChatTime** (Ye et al., 2025): "When LLM Meets Time Series: Can LLMs Perform Multi-Step Time Series Reasoning and Inference" - directly addresses time series reasoning with LLMs, including multi-step reasoning capabilities.
- **ITFormer** (and related work): Methods for bridging time series and natural language for multi-task temporal-textual QA should be compared.
- **Time-R1** (Luo et al., 2025): While cited, the comparison could be more detailed, especially regarding reasoning frameworks.

**Why This Matters**:

- These works represent the current state-of-the-art in time series + LLM integration.
- Without proper comparison, it is unclear what novel contributions TIMEOMNI-1 provides beyond existing methods.
- The architectural choices (text-based vs. embedding-based) should be positioned relative to these alternatives.

### 5. Technical Details Omitted

**Missing Information**:

- Input preprocessing pipeline (normalization, scaling, handling of missing values)
- Tokenization strategy for numerical values (precision, formatting)
- How task-specific prompts are constructed
- Details of the reward function design (mentioned but not fully specified in main text)

**Questions:**

## Questions

1. **Architecture Clarification**: Can the authors provide a clear description of how time series data is encoded and input to the model? Specifically:

   - Is numerical data converted to text strings? If so, what is the exact format?
   - Are there any learnable embeddings or encoders for time series, or is it purely text-based?
   - How are multiple time series (e.g., Task 2) represented in the input prompt?
2. **Scalability Validation**:

   - Have the authors tested their approach on longer sequences (e.g., 500+ time steps)? What are the limitations?
   - Can the approach handle multivariate time series with 10+ dimensions? How would this be encoded?
   - What is the computational efficiency compared to embedding-based approaches (e.g., OpenTSLM)?
3. **Design Choice Justification**:

   - Why was text serialization chosen over learned embeddings? What are the trade-offs?
   - How does this approach compare to methods like Time-LLM or OpenTSLM in terms of efficiency and effectiveness for long sequences?
4. **Dataset Characteristics**:

   - What are the typical sequence lengths in TSR-Suite? What is the distribution?
   - Are there examples with sequences longer than 100 time steps? If not, why not?
   - Why were only univariate or paired sequences chosen? Were multivariate scenarios considered?
5. **Related Work Comparison**:

   - How does TIMEOMNI-1 compare with ChatTime, which also addresses multi-step time series reasoning?
   - What are the key differences from ITFormer and similar methods?
   - Why should text-based serialization be preferred over architectural innovations like those in OpenTSLM?

---

> ### Author Response · Authors · 2025-11-22
> **Response to Reviewer XR2U [Part 1]**
>
> Thank you for your constructive and detailed review. Below, we provide responses to each comment.
>
> > **W1-part1:** The paper fails to clearly describe how time series data is processed and encoded into the language model.
> >
> > **W5-part2:** Tokenization strategy for numerical values (precision, formatting)
> >
> > **Q1-part1:** Can the authors provide a clear description of how time series data is encoded and input to the model? Specifically:
> >
> > - Is numerical data converted to text strings? If so, what is the exact format?
> > - Are there any learnable embeddings or encoders for time series, or is it purely text-based?
>
> Many thanks for your insightful question. We directly **tokenize the time series as text** and feed the resulting textual representation into the LLM. It matches what you described as “*numerical data converted to text strings*,” and we do not introduce learnable time series encoder. We have clarified this in $\underline{\text{Appendix F of revised paper}}$. In our manuscript, we do not claim innovation at the time series embedding or encoder level.

---

> ### Author Response · Authors · 2025-11-22
> **Response to Reviewer XR2U [Part 2]**
>
> > **W1-part2:** Comparison Baseline: The choice of text serialization vs. embedding-based approaches (as used in OpenTSLM, ChatTS, etc.) should be explicitly discussed.
> >
> > **W1-part6:**
> >
> > - **OpenTSLM** uses explicit architectural components (SoftPrompt or Flamingo-style Perceiver Resampler) for time series encoding.
> > - **Time-LLM** uses PatchTST-style tokenization with reprogramming layers.
> > - The current paper should justify why a simpler text-based approach is sufficient or preferred.
> >
> > **Q3-part1:** Why was text serialization chosen over learned embeddings? What are the trade-offs?
> >
> > **Q5-part3:** Why should text-based serialization be preferred over architectural innovations like those in OpenTSLM?
>
> We sincerely appreciate these comments. Below we present our experimental results and corresponding analyses.
>
> **(1) Comparative experiments: text-based vs. embedding-based approaches**
>
> TimeOmni-1 adopts a text-based strategy to represent time series inputs.
>
> For comparison, we implemented an embedding-based approach by **training a time series encoder from scratch** following the ChatTS architecture, which is conceptually aligned with OpenTSLM’s Soft Prompting and Time-LLM’s patch-based encoding. We use the same base model (Qwen2.5-Instruct-14B) with ChatTS to ensure a fair comparison. Results are summarized below.
>
> | | **Task1 ID** (ACC/SR) | **Task1 OOD** (ACC/SR) | **Task2 ID** (ACC/SR) | **Task2 OOD** (ACC/SR) | **Task3 ID** (MAE$\downarrow $/SR) | **Task3 OOD** (MAE$\downarrow $/SR) | **Task4 ID** (ACC/SR) | **Task4 OOD** (ACC/SR) |
> | - | - | - | - | - | - | - | -- | - |
> | Base Model (Qwen2.5-Instruct-14B) | 54.0/100.0 | 54.5/100.0 | 30.5/100.0 | 31.4/99.8 | 20.13/88.0 | **131.05**/64.1  | 35.1/100.0 | 34.8/100.0 |
> | Text-based (Ours) | 74.2/100.0 | 69.4/99.3 | **49.5**/100.0  | **47.9**/98.2  | **15.22**/90.2 | 139.44/84.9 | **38.9**/100.0 | **51.1**/99.0 |
> | Embedding-based | **88.5**/92.9 | **82.4**/97.7 | 40.3/68.7 | 38.7/62.9 | 22.15/75.1 | 162.23/58.4 | 18.2/59.5 | 20.0/51.9 |
>
>
> **(2) Discussion of using text serialization**
>
> 1. **Text-based strategy is a common and widely used approach.** There exist multiple viable strategies for encoding time series, including text-based (Time-R1, Time-MQA), learned embedding-based (ChatTS, OpenTSLM, Time-LLM), and image-based (TimeMaster) representations.
> 2. **Time series community currently lacks a universally accepted and pretrained time series encoder analogous to ViT [1] in vision domain.** In the absence of such general-purpose encoders, text-based input remains a stable and architecture-agnostic choice. However, designing a dedicated time series embedding module is a part of our future work.
> 3. **Our experiments show that a simple MLP-based encoder,** similar to those used in ChatTS, OpenTSLM, and Time-LLM, **does not significantly outperform our text serialization method.** More importantly, it noticeably reduces the success rate of producing valid responses, suggesting a negative impact on the base model’s instruction-following ability. A potential reason for the performance gap is the higher data requirement of the embedding-based method. Training a time series encoder from scratch is data-intensive, and our 2.3k CoT-Data might be inadequate.
> 4. **Encoder-based approaches introduce clear inference-efficiency drawbacks.** Since our Stage-2 training requires extensive sampling, adapt the same architecture with ChatTS significantly increases inference-time overhead, as demonstrated in our efficiency analysis for **W2-part3**.
> 5. **Current encoder-only methods mainly improve understanding but remain limited in generative reasoning.** Models such as ChatTS, OpenTSLM, and Time-LLM only add a time series encoder, while generation still relies entirely on the LLM’s next-token prediction. An ideally generative solution would require a dedicated decoder capable of reconstructing time series.
>
> Taken together, the text-based strategy is both effective and efficient for the settings considered in this work, where sequence lengths and input dimensionality is modest. Thank you again for raising this point. We have added the corresponding discussion to the revised paper in $\underline{\text{Appendix F.3: Discussion of Tokenizing Time Series into Text Inputs}}$.
>
> [1] *An image is worth 16x16 words: Transformers for image recognition at scale*, 2020.

---

> ### Author Response · Authors · 2025-11-22
> **Response to Reviewer XR2U [Part 3]**
>
> > **W1-part3:** How are long sequences handled? Are they truncated, sampled, or aggregated? Is there any preprocessing, normalization, or feature extraction?
> >
> > **W5-part1:** Input preprocessing pipeline (normalization, scaling, handling of missing values)
>
> Thank you for the helpful comments.
>
> **First, our tasks indeed involve long sequences.** The longest sequence in our dataset **contains 800 time steps**. We have added a new $\underline{\text{Appendix D.3: Statistics on Sequence Length and Token Budget}}$ that reports the maximum and average time series lengths used across all tasks. For your convenience, we also provide the statistics below:
>
> | | Task1 | Task2 | Task3 | Task4 |
> | - |  - | - | - | - |
> | MAX length / AVG length | **800** / 316.3 | 124 / 121.7 | 96 / 78.3 | 48 / 48.0 |
>
> **Second, regarding long-sequence handling and the “scaling” issue you mentioned**, we apply optimizations at both the data level and the training level:
>
> **(1) Data-level processing**
>
> - **Precision control.** We introduce a decimal-place reduction algorithm that limits each value to at most two decimal places while preserving the essential temporal information:
>
> $$
> x' =
> \\begin{cases}
> \\lfloor x \\rceil,
> & \\text{if } \\displaystyle \\frac{\\lvert x - \\lfloor x \\rceil \\rvert}{\\lvert x \\rvert} < 0.01, \\\\[6pt]
> \\text{rounded to at most two decimal places for } x,
> & \\text{otherwise}.
> \\end{cases}
> $$
>
>
>
>
> - **Resolution control**. For example, in Task 2, although the raw time series data is recorded at a 15-minute resolution, we adopt a 12-hour resolution, following the resolutions used in the original paper [1].
>
> **(2) Training-level processing**
> We set the maximum input sequence length to 8192 tokens during training, which is more than 8 times the average tokenized length (1106 tokens). This ensures that long sequences do not cause truncation.
>
> **Third, input preprocessing pipeline.** To retain the original data scale, we do not apply normalization. If any training sample contains missing values (e.g., NaN), we remove that sample entirely.
>
> [1] *CausalRivers -- Scaling up benchmarking of causal discovery for real-world time-series*, 2025.
>
> ---
>
> > **W1-part4:** What is the token budget for time series data vs. context?
> >
> > **W2-part1:** With text serialization, long sequences would consume substantial token budgets and face LLM context window limitations.
>
> Many thanks for pointing this out.  We compute the token budgets using the tokenizer of our base model, Qwen2.5-7B-Instruct. We have added a new $\underline{\text{Appendix D.3: Statistics on Sequence Length and Token Budget}}$ that reports the average token budgets used across all tasks. For your convenience, we also provide the statistics below:
>
> |   | Task1 | Task2 | Task3 | Task4 | Overall  |
> | -- | -  |  - | -  | -  | - |
> | **AVG token budget of input time series** $X$ | 1701  | 698   | 357   | 281   | 860   |
> | **AVG token budget of auxiliary context $C$** | 261   | 160   | 216   | 408   | 246  |
> | **AVG total token budget** | 1962  | 858   | 573   | 689   | **1106** |
>
> **Findings.** Long serialized sequences appear mainly in Task 1 and Task 2. We set the maximum input sequence length to 8192 tokens during training, which is more than 8 times the overall average token budget (1106 tokens).
>
> **We are far from hitting LLMs context-window limitations.** The average total token budget (1106 tokens) is significantly below the maximum supported input length of Qwen2.5-7B-Instruct (32,768 tokens). Thus, tokenizing time series into text inputs **does not** cause context-window limitations in this work.

---

> ### Author Response · Authors · 2025-11-22
> **Response to Reviewer XR2U [Part 4]**
>
> > **W1-part5:** How are multiple time series (Task 2) represented in the input?
> >
> > **Q1-part2:** How are multiple time series (e.g., Task 2) represented in the input prompt?
>
> Many thanks for pointing this out. **We represent multiple time series directly in text form.** Each series is serialized as a textual sequence, and the sequences are concatenated with clear separators in the input prompt. A full case study illustrating the exact input format is provided in $\underline{\text{Table 14 of Appendix L}}$.
>
> We compared our text-based representation with the alternative approach used in ChatTS, where multiple time series are arranged as a two-dimensional array and processed by a learned time series encoder.  We provide the results below:
>
> | | TASK2 ID (ACC/SR) | TASK2 OOD (ACC/SR) |
> | - | - | - |
> | Base Model (Qwen2.5-Instruct-14B) | 30.5/100.0 | 31.4/99.8 |
> | **Text-based representation** (Ours) | **49.5**/100.0 | **47.9**/98.2 |
> | **Two-dimensional array** (ChatTS) | 40.3/68.7 | 38.7/62.9 |
>
> **Finding.** Our results show that **the text-based representation does not underperform the two-dimensional array used in ChatTS.**
>
> ---
>
> > **W2-part1:**  Sequence Length Limitations:
> >
> > - All examples show short sequences (12-24 time steps).
> > - There is no evidence of handling longer sequences (hundreds or thousands of steps), which are common in real-world applications (e.g., hourly data over years).
> >
> > **Q4-part1:** Dataset Characteristics:
> >
> > - What are the typical sequence lengths in TSR-Suite? What is the distribution?
>
> Thanks for pointing this out. We clarify that **the average sequence lengths in all tasks are longer than 12–24 steps**. The maximum and average input lengths for each task are reported below:
>
> | | **Task1** | **Task2** | **Task3** | **Task4** |
> | -| - | - | - | - |
> | **MAX** length/**AVG** length | 800/**316.3** | 124/**121.7** | 96/**78.3** | 48/**48.0** |
>
> To further address the concern that “*no evidence of handling longer sequences (hundreds or thousands of steps)*” we evaluate our model on **TimeSeriesExam** benchmark, where each problem contains two time series of length 1024. This results in **a total input length of 2048 time steps**, substantially longer than 12–24 steps. **TimeOmni-1 continues to perform strongly under this setting**, as shown below:
>
> **TimeSeriesExam (Synthetic dataset).** Higher values indicate better performance.
>
> |  | Anomaly Detection | Causality Analysis | Noise Understanding | Pattern Recognition | Similarity Analysis | AVG|
> | -| -| -| -| -| -| -|
> | *Num of Samples* | *108* |  *72* | *84*| *362*| *120*| *—* |
> | **Base Model**| 37.0 | 44.4| 44.0| 50.0| **59.2**| 46.9 |
> | **Original Paper Result** (Phi-3.5) | 28.0 | **52.0**| 26.0| 47.0| 45.0| 39.6 |
> | **TimeOmni-1**| **42.6**| 37.5| **56.0**| **56.4**| 46.7| **47.8** |
>
> ---
>
> > **W2-part2:** Dimensionality Limitations:
> >
> > - The paper focuses on univariate time series (single-dimensional) or at most pairs of sequences.
> > - Real-world applications often involve multivariate signals (e.g., 12-lead ECG with 12 dimensions, sensor arrays with 50+ channels).
> > - No discussion of how high-dimensional multivariate time series would be handled.
>
> Thank you for pointing this out. To address your concern, we extend the dimension of Task 2 (Causality Discovery). We keep only two nodes causally related and add additional randomly selected nodes as distractors. The experimental results are as follows:
>
> | Input Dimensionality | 2 (Original Dimension) | 3| 4| 5|
> | - | -| -| - | -|
> | **TASK2 OOD (ACC/SR)** | 64.0/99.8| 61.9/99.1 | 62.4/99.3 | 60.4 / 98.3 |
>
> **Findings.** Expanding the input dimensionality by adding distractor nodes does not noticeably affect the model’s performance.
>
> To "*how high-dimensional multivariate time series would be handled*", **we represent multiple time series directly in text form.** Each series is serialized into a textual sequence. These sequences are then concatenated into the input prompt using clear separators, which are descriptive text outlining the attributes of each series.

---

> ### Author Response · Authors · 2025-11-22
> **Response to Reviewer XR2U [Part 5]**
>
> > **W2-part3:**
> >
> > - Text serialization of numerical data is token-inefficient compared to learned embeddings.
> >
> > - No efficiency/computation cost analysis comparing text-based vs. embedding-based approaches.
> >
> > **Q2-part3:** What is the computational efficiency compared to embedding-based approaches (e.g., OpenTSLM)?
> >
> > **Q3-part2:** How does this approach compare to methods like Time-LLM or OpenTSLM in terms of efficiency and effectiveness for long sequences?
>
> Many Thanks for your constructive comments. As suggested, we have adapted the encoder design from ChatTS as the **learned embedding baseline**. We conducted a comparative analysis using the same base model (Qwen2.5-Instruct-14B). The inference speed and **peak GPU memory** were evaluated under identical test conditions, with the key results summarized below:
>
> | | Sample Size | Avg Inference Speed per Sample (s) | Peak GPU Memory (GB) |
> | :- | :- | :- | :- |
> | **Text-based (Ours)** | 100 | 7.07 | 51.77 |
> | **Embedding-based**   | 100 | 22.01 | 32.04 |
>
> **Findings.**
>
> - **The embedding-based model exhibits 3x slower inference speed.** This is likely because the embedding-based architecture introduces an additional computational path through the time series encoder, and the complexity of this integrated structure limits our current implementation to inference modes that lack the kernel-level and scheduling optimizations found in vLLM.
> - **The text-based approach results in higher peak GPU memory usage.** It is important to note that we only report peak GPU memory here because average memory consumption is difficult to quantify precisely. As for the peak memory, the increased footprint in our method is mainly due to the KV cache allocated for long sequences (near 2k tokens). However, this **memory overhead remains manageable** and scales reasonably with input length. (**Note: In this test, the average time series length exceeds 300 points.**)
> - **Embedding-based approaches introduce clear inference-efficiency drawbacks.** Since our Stage-2 training involves extensive sampling, adopting such a design would substantially increase inference-time overhead.
>
> The current text-based serialization strategy is **sufficient and efficient for the tasks considered in this work**, which do not involve “Sequences with 1000+ time steps” and “Multivariate signals with 20+ dimensions”.
>
> However, we fully recognize the potential of embedding-based representations for handling more complex scenarios, and consider this as an important direction for our future research. Thanks again to your constructive comments. **We have added the corresponding discussion to the revised paper in $\underline{\text{Appendix F.3: Discussion of Tokenizing Time Series into Text Inputs}}$.**
>
> ---
>
> > **W2-part4:**  text serialization requires ~T tokens (one per value), whereas embedding-based approaches can compress this.
>
> Thank you for pointing this out. We agree with the reviewer that embedding-based approaches can compress the input. However,
>
> - **Text serialization is simple, direct, and performs better in our tasks.**
>    As shown in our ablations in $\underline{\text{(Appendix F.3)}}$, the text-based representation outperforms the learned embedding on most tasks. For general reasoning-style problems, feeding numerical data as text remains **effective and feasible**.
>
> - **Compression introduces information loss and unstable normalization.**
>   Even state-of-the-art compression techniques [1] do not show performance gains over non-compressed representations. Compression inherently removes information, and it remains unclear whether LLMs consistently interpret heavily normalized or rescaled time series.
> - **Embedding-based approaches largely relies on SFT, which may cause overfitting rather than genuine reasoning.** In contrast, our RL-based training improves generalization.
>
> [1] *DeepSeek-OCR: Contexts Optical Compression*, 2025.
>
> ---
>
> > **W2-part5:** The computational cost of processing long text sequences through LLMs is quadratic in sequence length.
> >
> > No discussion of context window limitations or how they were handled.
>
> Thank you for pointing this out. We acknowledge that LLM attention is quadratic in token length. However, **our token usage is far below the context window limitation.** The average total token budget in our tasks is **1,106 tokens**, while our base model (Qwen2.5-7B-Instruct) supports **32,768 tokens**. Thus, we are well within safe computational range. We have added a new $\underline{\text{Appendix D.3: Statistics on Sequence Length and Token Budget}}$ to discuss this.

---

> ### Author Response · Authors · 2025-11-22
> **Response to Reviewer XR2U [Part 6]**
>
> > **W3:** While TSR-Suite is substantial in sample size (23K+), the scope appears limited:
> >
> > **Short Sequences**: Examples suggest sequences are typically short (12-24 steps). Real-world time series reasoning often requires much longer histories.
> >
> > **Low Dimensionality**: Tasks focus on univariate sequences or pairs. High-dimensional multivariate scenarios are not addressed.
> >
> > **Limited Complexity**: The chosen tasks (scenario understanding, causality discovery) may not fully capture the complexity of reasoning over diverse time series modalities (e.g., sensor fusion, multi-scale temporal patterns).     2. diverse time series domain？
> >
> > **Missing Validation**: There is no evidence that the proposed approach would scale to:
> >
> > - Sequences with 1000+ time steps
> > - Multivariate signals with 20+ dimensions
> > - Complex sensor fusion scenarios
> >
> > **Q2-part1:** Have the authors tested their approach on longer sequences (e.g., 500+ time steps)? What are the limitations?
> >
> > **Q2-part2:** Can the approach handle multivariate time series with 10+ dimensions? How would this be encoded?
> >
> > **Q4-part2:**
> >
> > - Are there examples with sequences longer than 100 time steps? If not, why not?
> > - Why were only univariate or paired sequences chosen? Were multivariate scenarios considered?
>
> Many Thanks for your constructive comments.
>
> **(1) For “sequences are typically short (12-24 steps)”, "tested their approach on longer sequences (e.g., 500+ time steps)" and "Are there examples with sequences longer than 100 time steps?"**
>
> As clarified in our response to **W1**, the actual sequence lengths in TSR-Suite are **significantly longer**, with a **maximum of 800 steps in our dataset.**  We have also added detailed time series length statistics to $\underline{\text{Table 4 of Appendix D.3}}$.
>
> In addition, we further evaluate on the third-party TimeSeriesExam benchmark, where instance contains **two sequences of length 1024 (total input length 2048)**. TimeOmni-1 performs robustly in this long-sequence testbed.
>
> **(2) For “low dimensionality”, "multivariate signals with 20+ dimensions" and "Were multivariate scenarios considered?"**
>
> We do not claim to support 20+ dimensional inputs in this work. However, to directly address the reviewer’s concern, we extend Task 2 to **higher-dimensional inputs (up to 5 dimensions)**. The experimental results are as follows:
>
> | Input Dimensionality   | 2 (Original Dimension) | 3         | 4         | 5           |
> | ---------------------- | ---------------------- | --------- | --------- | ----------- |
> | **TASK2 OOD (ACC/SR)** | 64.0/99.8              | 61.9/99.1 | 62.4/99.3 | 60.4 / 98.3 |
>
> **Findings.** Expanding the input dimensionality by adding distractor nodes show **no significant performance degradation**.
>
> **(3) For “limited complexity” “diverse time series domains,” and “sensor fusion scenarios”**
>
> We additionally include results on **three external benchmarks (cross 13 tasks, 3,406 samples)** to **strengthen the evaluation breadth**, please refer to  $\underline{\text{Appendix K: Additional Evaluation on External Time Series QA Benchmarks}}$ or the response to **W1 of Reviewer: YkbD**. The results indicate that modeling time series reasoning as three core capabilities (perception, extrapolation, and decision-making) generalizes well across domains. In particular, TimeOmni-1 shows strong performance on **MTBench** benchmark [1], where **fusing textual context** with **time series** is especially important for effective reasoning.
>
> [1] *MTBench: A Multimodal Time Series Benchmark for Temporal Reasoning and Question Answering*, 2025.

---

> ### Author Response · Authors · 2025-11-22
> **Response to Reviewer XR2U [Part 7]**
>
> > **W4: Insufficient Related Work Coverage**
> > Several important recent works on time series QA and reasoning are not adequately discussed:
> >
> > **Missing Works**:
> >
> > - **ChatTime**: directly addresses time series reasoning with LLMs, including multi-step reasoning capabilities.
> > - **ITFormer**: Methods for bridging time series and natural language for multi-task temporal-textual QA should be compared.
> > - **Time-R1**: While cited, the comparison could be more detailed, especially regarding reasoning frameworks.
> >
> > **Q5-part1:** How does TIMEOMNI-1 compare with ChatTime, which also addresses multi-step time series reasoning?
> >
> > **Q5-part2:** What are the key differences from ITFormer and similar methods?
>
> Thank you for the insightful comments.
>
> **We have revised $\underline{\text{Section 2 Related Work}}$ to include** ChatTime (Ye et al., 2025) and  ITFormer (and related work) as related work.
>
> To further clarify the positioning of TimeOmni-1, we supplement comparisons with **ChatTime, ITFormer, OpenTSLM, Time-R1**, evaluated under the same settings as our benchmark. The results as below:
>
> | |Task1 ID (ACC/SR)|Task1 OOD (ACC/SR)|Task2 ID (ACC/SR)|Task2 OOD (ACC/SR)|Task3 ID (MAE$\downarrow$/SR)|Task3 OOD (MAE$\downarrow$/SR)|Task4 ID (ACC/SR)|Task4 OOD (ACC/SR)|
> |-|-|-|-|-|-|-|-|-|
> |*Time Series Language Model*|  |  | | | | |  | |
> |**ChatTime-7B-Chat**|18.2 / 11.0|29.8 / 12.7|- / -|- / -|14.47 / 100|154.55 / 100|- / 0.0|- / 0.0|
> |**ITFormer-7B**|43.8 / 100|47.5 / 100|15.0 / 47.0|14.6 / 42.0|29.55 / 96.0|230.04 / 100.0|25.0 / 100|41.7 / 100|
> |**OpenTSLM-llama-3.2-3b-ecg-flamingo**|- / 5.0|- / 3.2|1.6 / 23.0|3.3 / 26.5|- / 0.2|- / 0.0|17.8 / 98.4|16.2 / 98.9|
> |*Time Series Reasoning Model*| | | | | | | | |
> |**Time-R1**|30.9 / 94.0|34.0 / 92.5|30.2 / 53.8|31.4 / 48.9|17.61 / 38.7|- / 6.3|27.8 / 95.7|32.2 / 93.1|
> |**TimeOmni-1**|**90.7** / 97.5|**87.7** / 98.3|**69.3** / 99.8|**64.0** / 99.8|14.30 / 93.8|**145.53** / 82.3|**47.9** / 100.0|**58.9** / 100.0|
>
> **Findings:**
>
> **(1) Time Series Language Model**
>
> - **ChatTime-7B-Chat does not natively support multivariate input**, hence it yields no results for Task 2. Its extensive instruction fine-tuning for forecasting tasks appears to have compromised its instruction-following capability on other tasks, leading to poor performance on Task 1 and Task 4. However, its tokenizer-de-tokenizer design grants it superior numerical representation, enabling a 100% success rate (SR) in generating valid sequence-outputs (Task 3).
> - **OpenTSLM-llama-3.2-3b-ecg-flamingo** demonstrates very weak instruction-following capability, suggesting that relying heavily on instruction fine-tuning can diminish the base model’s inherent general abilities.
> - **ITFormer-7B** performs poorly overall yet shows a high success rate (SR). Upon inspection, we found that this is primarily attributable to its use of the same $\\langle answer \\rangle$ tags as TIMEOMNI-1, which standardizes output formatting.
> - Both **OpenTSLM-llama-3.2-3b-ecg-flamingo** and **ITFormer-7B** utilize patch encoder, which require the input time series length to be an integer multiple of the patch length. This design limits their flexibility, as they cannot handle inputs of arbitrary length (e.g., ITFormer-7B's default patch length is $60$, requiring $ \\text{input\\_length} \bmod 60 = 0 $
>  to avoid errors).
>
> **(2) Time Series Reasoning Model**
>
> - **Time-R1 and TimeOmni-1**, as reasoning models, share a similar underlying architecture with DeepSeek-R1 [1]. **Our focus is not on proposing a new reasoning framework, and we do not claim novelty in that regard.** Instead, our contribution lies in providing the first systematic guideline and key insights for developing time series reasoning models.
>
> **Positioning of TimeOmni-1:**
>
> **(1) Compared to *Time Series Language Models* (ChatTime, ITFormer, OpenTSLM)**, TimeOmni-1 offers two key advantages:
>
> - **A strong and general-purpose checkpoint.** As the **first generalized reasoning model for time series data**, TimeOmni-1 provides a foundation for building more advanced time series AI.
> - **Broader capability and greater data diversity.** TimeOmni-1 is trained on three core capabilities: **perception, extrapolation, and decision-making** using the proposed TSR-Suite dataset. Additional evaluations across **13 tasks and 3,406 samples** show that its learned reasoning abilities generalize across domains and tasks.
>
> **(2) Compared to the *Time Series Reasoning Model* (Time-R1)**:
>
> - **TimeOmni-1 beyond merely mirroring DeepSeek-R1** [1] by developing systematic time series reasoning capabilities and offering practical insights for constructing time series reasoning models.
> - **The evidence for the joint training paradigm "is convincing"** (as highlighted by **Reviewer ruC5**), demonstrating the feasibility of building a general purpose time series reasoning model.
>
> [1] *Deepseek-r1: Incentivizing reasoning capability in llms via reinforcement learning*, 2025.

---

> ### Author Response · Authors · 2025-11-22
> **Response to Reviewer XR2U [Part 8]**
>
> > **W5-part3:** How task-specific prompts are constructed
>
> Thank you for raising this question. Our task-specific prompts are constructed in three components:
>
> - **Task Format.** Each task has a well-defined prompt, mainly contains (1) observed time series inputs $X$, (2) auxiliary context $C$ (e.g., task instructions or external knowledge). We include 4 complete case studies in $\underline{\text{Appendix L}}$ to illustrate the specific prompt. The generated QA-pairs are **reviewed by human using our human evaluation interface in $\underline{\text{Appendix D.5}}$** to ensure they satisfy the proposed design principle "*Principle 2 — QA pairs must ensure context sufficiency*".
>
> - **Prompt for Hierarchical CoT Annotation.** As part of our human-guided annotation pipeline, human experts use structured templates to elicit consistent reasoning traces from both the LLM Analyzer and the LLM Rewriter. The full prompts used for all four tasks are provided in $\underline{\text{Appendix E.1}}$.
> - **System prompt.** Depending on the task type, we use different system prompts for discrete-output tasks and sequence-output tasks. The full system-prompt templates are also included in $\underline{\text{Appendix E.2}}$.
>
> Besides, we further investigate the robustness of TimeOmni-1 to prompt variations. Please refer to the newly added $\underline{\text{Appendix E.3: Model Robustness to Prompt Variations}}$ or the response to **W5 of Reviewer: 2VxW**.
>
> ---
>
> > **W5-part4:** Details of the reward function design (mentioned but not fully specified in main text)
>
> Thanks for pointing this out. The full details of the reward function design are provided in $\underline{\text{Appendix G}}$.
>
> In addition, we conduct a thorough sensitivity analysis of the reward design. Please refer to  the newly added $\underline{\text{Appendix G.1: Reward Sensitivity Analysis}}$ or our response to **W4 of Reviewer: 2VxW** for these results.
>
> ---
> We once again thank you for taking the time to review our paper and for your insightful and constructive comments.
> We truly appreciate this opportunity to clarify our work and would be grateful for any further feedback.
>
> Best regards,
>
> The Authors

---

> > ### Comment · Reviewer_XR2U · 2025-11-27
> > **Response to Rebuttal**
> >
> > Thank you for the detailed responses and the additional experiments.
> >
> > The rebuttal has successfully addressed my primary concerns, specifically regarding the justification for the text serialization strategy versus learned embeddings, as well as the handling of long sequences and computational efficiency. The new experimental results provide solid evidence for the design choices made in the paper.
> >
> > I believe this work makes a solid contribution to the field of Time Series Reasoning. Consequently, I have raised my rating to 8: Accept.

---

> > > ### Author Response · Authors · 2025-11-27
> > > **Sincere Gratitude from Authors**
> > >
> > > Dear Reviewer,
> > >
> > > We are very glad to hear that our rebuttal has addressed your concerns. Thank you again for the detailed and constructive comments, which have helped us further strengthen this work.
> > >
> > > We are also deeply grateful for your strong support and positive evaluation, which are extremely important and encouraging to us.
> > >
> > > Best regards,
> > >
> > > The Authors

---

> ### Author Response · Authors · 2025-11-26
> **Request of Reviewer's attention and feedback**
>
> Dear Reviewer,
>
> As a gentle reminder, it has been more than four days since we submitted our rebuttal, and we were wondering whether our response has addressed your concerns.
> We look forward to your feedback and are prepared to address any further concerns you may have.
>
> Thank you again for your valuable and constructive feedback, which has inspired further improvements to our paper.
>
> Best regards,
>
> The Authors

---

### Official Review · Reviewer_YkbD · 2025-10-31

**Soundness:** 1
**Presentation:** 3
**Contribution:** 2
**Rating:** 4
**Confidence:** 3

**Summary:**

The paper presents a reasoning model for time series data. The authors develop a new dataset by augmenting existing time series data and train a model with SFT and RL to conduct multistep reasoning on a variety of tasks. The results show that this prior enables successful training and increased performance on the proposed dataset.

**Strengths:**

- It's true that the majority of prior work focused more on evaluating models than training them to be better at these tasks
- The ablations are appropriate and useful for understanding how the model works
- The human-generated reasoning traces are a valuable contribution to the field

**Weaknesses:**

A serious weakness of the paper is that the authors only evaluate on the dataset they propose. It's not particularly surprising that training on a dataset improves in-distribution performance. I'd be more impressed if the authors showed that their model improves performance on third party benchmarks (including those they reference), or that it mitigates this "overthinking" problem that they motivate their dataset with.

The evaluation section and tables mention an "OOD" dataset, but it's not clear how this is constructed or in what sense it is out of distribution.

**Questions:**

- "It remains unclear which tasks genuinely demand reasoning capabilities over time series." I'm not sure what this means. Several recent works (including those you cite like Merrill et al 2024 and Cow et al. 2024) define tasks that require reasoning over time series.
- Line 157: "Many questions are overly simple and straightforward, where invoking reasoning leads to over-thinking" and Line 185 "QA-pairs must reward reasoning". Why is it a problem that reasoning doesn't provide an increase in performance? Is this really an issue with the benchmark, or does it say more about the models themselves?  If the models are overthinking a reasonable, realistic question then my instinct would be to improve the models, not find some set of questions where this wasn't the case.

---

> ### Author Response · Authors · 2025-11-22
> **Response to Reviewer YkbD [Part 1]**
>
> We would like to sincerely thank Reviewer YkbD for providing the insightful review and comments.
>
> > **W1**: It's not particularly surprising that training on a dataset improves in-distribution performance. I'd be more impressed if the authors showed that their model improves performance on third party benchmarks.
>
> We sincerely appreciate for raising this important point.
>
> **(1) Beyond In-Distribution Performance**
>
> Although our main experiments are conducted on the proposed TSR-Suite dataset, **all tasks in $\underline{\text{Table 1}}$ include out-of-distribution (OOD) evaluations**. We fully agree that in-distribution gains are expected, which is precisely why our evaluation emphasizes OOD generalization beyond the in-distribution setting.
>
> **(2) Performance on Three Additional Time Series QA Benchmarks (cross 13 tasks, 3,406 samples)**
>
> To fully address your concern, we conducted additional experiments on **three external time series QA benchmarks**:
>
> - **MTBench** [1] (real-world time series, 4 tasks, 2,380 samples),
> - **TimeSeriesExam** [2] (synthetic time series, 5 tasks, 746 samples),
> - **CaTS-Bench** [3] (real-world time series, 4 tasks, 280 samples).
>
> All evaluations include **two baselines**:
>
> - **Qwen2.5-7B-Instruct**, which serves as the base model of TimeOmni-1,
> - **The original results** reported by each benchmark **for fair comparison**.
>
> All results have been added to the $\underline{\text{Appendix K of Revised Paper}}$. For your convenience, we summarize the results below, where **higher values indicate better performance**.
>
> **MTBench (Real-world dataset)**
>
> |      | FinanceQA (7-Day) | WeatherQA (7-Day) | FinanceQA (30-Day) | WeatherQA (14-Day) | AVG      |
> | -| -| -| -| -| --|
> | *Num of Samples*    | *484*        | *666*           | *523*   | *707*       | *—*      |
> | **Base Model**   | 62.2              | 44.7        | 53.5        | 36.2        | 49.2     |
> | **Original Paper Result**  (GPT-4o) | 65.1        | 41.7              | 52.8               | **44.8**           | 51.1     |
> | **TimeOmni-1**       | **72.1**     | **51.4**          | **66.2**           | 42.6               | **58.1** |
>
> **TimeSeriesExam (Synthetic dataset)**
>
> |              | Anomaly Detection | Causality Analysis | Noise Understanding | Pattern Recognition | Similarity Analysis | AVG      |
> | -| -|- | -| -| -|- |
> | *Num of Samples* | *108*| *72* | *84* | *362* | *120*| *—*      |
> | **Base Model**   | 37.0    | 44.4       | 44.0                | 50.0                | **59.2**            | 46.9     |
> | **Original Paper Result** (Phi-3.5) | 28.0              | **52.0**           | 26.0                | 47.0                | 45.0                | 39.6     |
> | **TimeOmni-1**          | **42.6**          | 37.5               | **56.0**            | **56.4**            | 46.7                | **47.8** |
>
> **CaTS-Bench (Real-world dataset)**
>
> |                                                | Caption Retrieval | TimeSeries Retrieval | Amplitude Comparison | Mean Comparison | AVG      |
> | -| -|- |- |- |- |
> | *Num of Samples*                               | *100*             | *100*                | *40*                 | *40*            | *—*      |
> | **Base Model**                                 | **69.0**          | **46.0**             | 87.5                 | 57.5            | **65.0** |
> | **Original Paper Result** (LLaVA v1.6 Mistral) | 44.0              | 29.0                 | 43.0                 | 35.0            | 37.8     |
> | **TimeOmni-1**                                 | 57.0              | 37.0                 | **92.5**             | **67.5**        | 63.5     |
>
> **Findings:**
>
> **(1) TimeOmni-1 achieves the strongest gains on MTBench.** MTBench is a multimodal time series QA setting where the auxiliary text ($C$) contains task-relevant information (e.g. external events), **matching our TSR-Suite formulation:** $(R, y) \sim p_\theta(R, y \mid X, C) = p_\theta(R \mid X, C), p_\theta(y \mid R, X, C),$ meaning both the reasoning path ($R$) and final answer ($y$) depend on the interaction between the time series input ($X$) and contextual information ($C$). This alignment explains why TimeOmni-1 even surpasses the originally reported GPT-4o performance.
>
> **(2) TimeOmni-1 perform well on TimeSeriesExam and CaTS-Bench.** In these two benchmarks, the text serves mainly as task instructions rather than auxiliary information, so the reasoning relies almost entirely on the time series modality, which is **less aligned with the multimodal reasoning scope defined in TSR-Suite.**
>
> **Overall, TimeOmni-1’s strong performance across three heterogeneous benchmarks demonstrates that its learned time series reasoning capabilities generalize across domains and tasks.**
>
> [1] *MTBench: A Multimodal Time Series Benchmark for Temporal Reasoning and Question Answering*, 2025.
>
> [2] *TimeSeriesExam: A time series understanding exam*, 2024.
>
> [3] *CaTS-Bench: Can Language Models Describe Numeric Time Series?*, 2025.

---

> ### Author Response · Authors · 2025-11-22
> **Response to Reviewer YkbD [Part 2]**
>
> > **W2:** The evaluation section and tables mention an "OOD" dataset, but it's not clear how this is constructed or in what sense it is out of distribution.
>
> Many thanks for pointing this out. TSR-Suite constructs out of distribution (OOD) datasets via **domain**, **geographical**, and **building-level distribution shifts**. The full construction details and statistics of all OOD datasets are provided in $\underline{\text{Appendix D.1}}$ and $\underline{\text{Appendix D.2}}$. For your convenience, we summarize the key details below.
>
> **Task 1: Scenario Understanding.** OOD is defined by **domain shift**. OOD samples come from Agriculture, Education, and Healthcare domains, which are *different from the training domains*.
>
> **Task 2: Causality Discovery.** All data come from hydrology domain. OOD is defined by **geographical shift**: ID samples come from rivers in *Eastern Germany*, while OOD samples come from *Bavaria*, a different region with distinct hydrological characteristics.
>
> **Task 3: Event-aware Forecasting.** OOD is defined by **domain shift**: ID samples use human mobility time series (*New York City taxi drop-offs with aligned events*), while the OOD test set uses EWELD (*electricity load time series with weather events*), which differs in scale and event–series interactions.
>
> **Task 4: Decision Making.** All data come from CityLearn [1]. OOD is defined by **building-level distribution shift**: the OOD test set uses two buildings whose load profiles *exhibit distinct temporal patterns compared to those in the training set.*
>
> In summary, TSR-Suite constructs OOD dataset via **domain**, **geographical**, and **building-level distribution shifts**, ensuring that the underlying temporal dynamics deviate from the training distribution.
>
> [1] *The citylearn challenge 2022: Overview, results, and lessons learned.*, 2023.

---

> ### Author Response · Authors · 2025-11-22
> **Response to Reviewer YkbD [Part 3]**
>
> > **Q1:** "It remains unclear which tasks genuinely demand reasoning capabilities over time series." I'm not sure what this means. Several recent works (including those you cite like Merrill et al 2024 and Cow et al. 2024) define tasks that require reasoning over time series.
>
> Many thanks for pointing this out. We agree that several recent works include tasks involving reasoning over time series. However, after surveying **19** up-to-date studies, we found **substantial heterogeneity in task formulations, reasoning requirements, and data assumptions**. We conclude that the field still **lacks a clear definition or boundary for what constitutes "time series reasoning tasks".**
>
> This motivates our contribution in TSR-Suite. We systematically formalize reasoning-critical time series tasks by grounding them in three core capabilities: **perception, extrapolation, and decision-making**. When constructing QA pairs, we propose two principles: **ensuring QA requires reasoning and context sufficiency.** Reviewer XR2U and Reviewer 2VxW both highlighted this as a strength:
>
> > Reviewer XR2U: “*Clear Problem Identification: The analysis of limitations in existing datasets (Figure 1) is **well-executed** and **provides strong motivation for the proposed work.***”
> >
> > Reviewer 2VxW: *“The paper **tackles time series as a reasoning problem** and operationalizes this view with a benchmark that spans perception, extrapolation, and decision making. **This scope is broader** than pure forecasting and **motivates models that can explain and decide**.”*
>
> In summary, although TSR-Suite remains a preliminary effort, we hope it provides the first systematic guideline and useful insights for the design and development of reasoning-critical time series tasks and time series reasoning models.
>
> ---
>
> > **Q2:** "Many questions are overly simple and straightforward, where invoking reasoning leads to over-thinking" and "QA-pairs must reward reasoning". Why is it a problem that reasoning doesn't provide an increase in performance? Is this really an issue with the benchmark, or does it say more about the models themselves? **If the models are overthinking** a reasonable, realistic question then **my instinct would be to improve the models**, not find some set of questions where this wasn't the case.
>
> Thank you for your insightful and constructive comments.
>
> **(1) We fully agree that improving the models is the natural intuition and also the ultimate goal for addressing overthinking.**
> We view the development of models that can adaptively decide when to reason and when not to reason as a promising direction, and we include this in our future scope. Similar ideas have already been explored in text and math reasoning [1-2].
>
> **(2) Improving models and improving datasets are not contradictory. Better datasets are prerequisites for better models.**
> As shown in our analysis of limitations in existing time series QA datasets $\underline{\text{(Figure.1 and Appendix C)}}$, without reasoning already achieve optimal performance, which means explicit reasoning neither helps nor is required. However, if we aim to build models with stronger reasoning capabilities, **the tasks themselves must require reasoning rather than encourage pseudo or redundant reasoning on overly simple problems**. In this sense, strengthening the benchmark is a necessary step toward meaningful improvements in time series reasoning capabilities.
>
> [1] *Think When You Need: Self-Adaptive Chain-of-Thought Learning*, 2025.
>
> [2] *When to Continue Thinking: Adaptive Thinking Mode Switching for Efficient Reasoning*, 2025.
>
> ---
> We once again thank you for taking the time to review our paper and for your insightful and constructive comments.
> We truly appreciate this opportunity to clarify our work and would be grateful for any further feedback.
>
> Best regards,
>
> The Authors

---

> ### Comment · Reviewer_YkbD · 2025-11-24
>
> These are thoughtful and rigorous experiments so I'll bump my rating two points.

---

> ### Author Response · Authors · 2025-11-25
> **Sincere Gratitude from Authors**
>
> We are truly delighted that our responses have addressed your concerns. We would like to once again express our sincere gratitude for taking the time to review our paper and for providing such insightful and invaluable comments.

---

### Author Response · Authors · 2025-11-22
**Response to all reviewers & Summary of revisions**

We sincerely thank all the reviewers for their insightful comments and valuable suggestions, which have greatly helped us improve our paper.

This paper introduces the **Time Series Reasoning Suite (TSR-Suite)**, the first comprehensive time series reasoning suite containing over 23K samples. It formalizes four tasks spanning three core reasoning capabilities: perception, extrapolation, and decision-making. Based on this, we present **TimeOmni-1**, the first generalized reasoning model for time series data. We provide a comprehensive evaluation and **four key findings**, which we hope will serve as the first systematic guideline for the development of time series reasoning models.

The reviewers expressed positive views on our work, describing the proposed TSR-Suite and TimeOmni-1 as **"a substantial contribution"**, with "**well-executed and provides strong motivation**" and **"innovative"**. The writing was noted as **"easy to follow and very clear in structure"**. Our experiments were considered **"reveal insights that are very useful for the community"**, with particular emphasis on the **"evidence for cross task transfer is convincing"**.

The reviewers also provided many insightful comments. We have made every effort to address all the concerns. $\underline{\text{Below is a summary of the major revisions:}}$

- **TimeOmni-1’s performance on third party time series reasoning benchmarks (Reviewer YkbD *&* Reviewer ruC5)**: We conducted additional experiments on **3 external time series QA benchmarks**, covering **13 tasks** and **3,406 samples**. The results demonstrate that TimeOmni-1 has learned time series reasoning capabilities that generalize across domains and tasks.

  $\text{Newly added} \rightarrow \underline{\text{Appendix K: Additional Evaluation on External Time Series QA Benchmarks}}$
- **Clarification on time series encoding (Reviewer XR2U)**: (1) We clarified that we directly tokenize the time series as text. (2) We have reproduced and evaluated the embedding-based strategy. Under the setting of this work, tokenizing time series as text outperformed retraining a time series encoder in both performance and inference efficiency, with only a slight disadvantage in memory usage. (3) We note that text-based strategy is a common approach, and we do not claim innovation at the time series embedding level.

  $\text{Newly added} \rightarrow \underline{\text{Appendix F.3: Discussion of Tokenizing Time Series}}$
- **Additional reporting on sequence length, token budget, and dimensionality (Reviewer XR2U)**: (1) In TSR-Suite, Task 1 has a maximum input length of 800 time steps, with an average length of 316.3; the average input series length for each task exceeds 48 time steps. (2) The average total token budget (1,106 tokens) is significantly below the input length limit (32,768 tokens) of our base model Qwen2.5-7B-Instruct. (3) While we do not claim able to handle “*multivariate signals with 20+ dimensions*,” we have added experiments expanding Task 2 inputs to 5 dimensions.

  $\text{Newly added} \rightarrow \underline{\text{Appendix D.3: Statistics on Sequence Length and Token Budget}}$
- **Expanded comparisons with ChatTime, ITFormer, and OpenTSLM (Reviewer XR2U)**: We provide additional comparisons against these models, analyzing key differences and the positioning of TimeOmni-1.

- **Confidence intervals and full RL training curves (Reviewer 2VxW)**: We report 95% confidence intervals (over 5 runs) and provide the full RL training dynamics.

  $\text{Newly added} \rightarrow \underline{\text{Appendix I: Reinforcement Learning Training Dynamics}}$
- **Reward sensitivity and prompt variation analysis (Reviewer 2VxW)**: (1) We conducted a reward sensitivity analysis by sweeping key reward components ($\alpha$, $R_{\text{count}}$, $\lambda$) and evaluating their impact on training stability, consuming **130 A100 GPU hours**. (2) We designed three types of prompt perturbations (applied to questions and system prompts), showing that TimeOmni-1 is highly robust to prompt variations.

  $\text{Newly added} \rightarrow \underline{\text{Appendix G.1: Reward Sensitivity Analysis}} \text{ and } \underline{\text{Appendix E.3: Model Robustness to Prompt Variations}}$

- **Scaling of Training Dataset (Reviewer 2VxW)**: We conducted extensive scaling experiments not only for **Stage 1 (SFT)** but also for **Stage 2 (RL)**, thereby evaluating both the quantity and the impact of CoT-Data and RL-Data. These experiments consumed **360 A100 GPU hours** in total.

  $\text{Newly added} \rightarrow \underline{\text{Appendix J: Scaling of Training Dataset}}$

**After a full 8 days of experiments, we have added over 200 new experimental results to address the above concerns.** The revisions are highlighted in $\underline{\text{orange.}}$

The reviewers’ valuable suggestions have been extremely helpful in improving our paper. We are happy to answer any further questions.

We look forward to your feedback.

---

### Author Response · Authors · 2025-12-01
**Summary of Rebuttal Consensus (Pre-Incident Scores: 6, 8, 6, 8; Average 7.0)**

Dear AC, SAC, and PC,

We understand that you are taking over under difficult circumstances due to the OpenReview bug. **The bug was reported at *10:09am EST (US) on November 27*.** To support your assessment, we briefly summarize the status of our paper as it was before the report time.

All information below refers only to reviews and score updates made **at least 9 hours before** the report time.

---

**(1) Score Changes**

Three reviewers had already **increased their scores**, bringing our paper to **6, 8, 6, 8** (**Average 7.0**). The detailed timing and score changes history are summarized below:

***Table. Reviewers’ scores before/after rebuttal as of 9 hours before the OpenReview bug was reported.***

|Reviewer|Initial $\rightarrow$ After Rebuttal|Time of Last Score Update|Time Relative to Bug Report|Score Changes History Link|
|-|-|-|-|-|
|***YkbD***|**4 $\rightarrow$ 6**|3:52pm EST, Nov 24|**3 days** before bug report|https://openreview.net/revisions?id=DAl8CTeErJ|
|***XR2U***|**4 $\rightarrow$ 8**|1:18am EST, Nov 27|**9 hours** before bug report|https://openreview.net/revisions?id=rMb5wawXqw|
|***2VxW***|**6 $\rightarrow$ no response so far**| -| -|- |
|***ruC5***|**6 $\rightarrow$ 8**|7:05pm EST, Nov 26|**1 day** before bug report|https://openreview.net/revisions?id=KmGyOFIvtp|
|**Average Rating**|**5.0 $\rightarrow$ 7.0**| -| -| -|


---


**(2) Reviewers’ Feedback on Our Rebuttal**

**Reviewer ```YkbD``` (score: 4 $\rightarrow$ 6)**

- Responded *3 days before the bug was reported*, and the concerns were **fully addressed**:

  > **Reviewer YkbD**: "*These are thoughtful and rigorous experiments so **I'll bump my rating two points.***"

**Reviewer ```XR2U``` (score: 4 $\rightarrow$ 8)**

- Responded *9 hours before the bug was reported*, and the concerns were **fully addressed**:

  >  **Reviewer XR2U**: "*Thank you for the detailed responses and the additional experiments.*
  >
  >  ***The rebuttal has successfully addressed my primary concerns,** specifically regarding the justification for the text serialization strategy versus learned embeddings, as well as the handling of long sequences and computational efficiency. The new experimental results provide solid evidence for the design choices made in the paper.*
  >
  >  *I believe this work makes **a solid contribution to the field of Time Series Reasoning.** Consequently, **I have raised my rating to 8: Accept.***"

**Reviewer ```ruC5``` (score: 6 $\rightarrow$ 8)**

- Responded *1 day before the bug was reported*, and the concerns were **fully addressed**:

  > **Reviewer ruC5**: "*Thank you for the response! **I remain positive** regarding my feedback for this paper.*"
  >
  > **Reviewer ruC5** : "***I did update my score with my last response** in case you haven't noticed.*"

---

> ### Author Response · Authors · 2025-12-01
> **Summary of Rebuttal Consensus (Pre-Incident Scores: 6, 8, 6, 8; Average 7.0) – Continued**
>
> **Reviewer ```2VxW``` (score: 6 $\rightarrow$ no response so far)**
>
> Originally gave a **positive score of 6**, with **Soundness: 3** (good), **Presentation: 3** (good), **Contribution: 3** (good). Although Reviewer 2VxW did not respond to our rebuttal before the bug was reported, we thoroughly addressed all concerns raised:
>
> 1. **A contribution map that separates what is new (W1)**:  We have created a contribution map to clearly delineate what our work adds to the community.
>
> 2. **Scaling of Training Dataset (W2 & Q3)**: We conducted extensive scaling experiments not only for **Stage 1 (SFT)** but also for **Stage 2 (RL)**, thereby evaluating both the quantity and the impact of CoT-Data and RL-Data. These experiments consumed **360 A100 GPU hours** in total.
>
>    $\text{Newly added} \rightarrow \underline{\text{Appendix J: Scaling of Training Dataset}}$
>
> 3. **Confidence intervals (W3)**: We report 95% confidence intervals (over 5 runs).
>
> 4. **Reward sensitivity (W4 & Q2)**: We conducted a reward sensitivity analysis by sweeping key reward components ($\alpha$, $R_{\text{count}}$, $\lambda$) and evaluating their impact on training stability, consuming **130 A100 GPU hours**.
>
>    $\text{Newly added} \rightarrow \underline{\text{Appendix G.1: Reward Sensitivity Analysis}}$
>
> 5. **Prompt variation analysis (W5)**: We designed three types of prompt perturbations (applied to questions and system prompts), showing that TimeOmni-1 is highly robust to prompt variations.
>
>    $\text{Newly added} \rightarrow \underline{\text{Appendix E.3: Model Robustness to Prompt Variations}}$
>
> 6. **Full RL training curves (Q1)**: We provide the full RL training dynamics.
>
>    $\text{Newly added} \rightarrow \underline{\text{Appendix I: Reinforcement Learning Training Dynamics}}$
>
> 7. **Comparisons to recent live benchmarks (FutureX) and interactive benchmark (TSAIA) (W6 & Q4)**: (1) The mentioned live benchmarks (FutureX) are outside the scope of our model. **TimeOmni-1 is designed for time series reasoning rather than event prediction.** (2) We evaluate our model on the interactive benchmark (TSAIA). **TimeOmni-1 achieves surprisingly strong performance on the TSAIA benchmark.** Even though TimeOmni-1 is not trained on coding or tool-calling data, it still achieves a 6.5% improvement over the base model ($38.5 \rightarrow 45.0$).
>
> - **In total,** we added **4 new appendix sections** and **over 150 new experimental results** to address Reviewer 2VxW’s concerns as thoroughly as possible. Although Reviewer 2VxW has not yet responded to these additions, we believe they **can effectively address** the key concerns raised.
>
> ---
>
> **(3) Final Remarks**
>
> We would like to affirm that the discussion of our paper complied with the rules of anonymous peer review, and that we did not engage in any deanonymization, off-platform contact with reviewers, or collusion.
>
> In summary, **before the incident** our paper had reached a **consensus average score of 7.0 (6, 8, 6, 8)**. We had received **very positive responses from three reviewers (YkbD, XR2U, ruC5)**, and we have made substantial, concrete improvements to address the concerns of **Reviewer 2VxW**.
>
> As **Reviewer XR2U** stated, “***I believe this work makes a solid contribution to the field of Time Series Reasoning**.*” We sincerely hope that you will evaluate our paper in light of this pre-incident consensus (**scores:** **6, 8, 6, 8**) and the substantial improvements we have made in response to all reviewers.
>
> We sincerely appreciate your time and the additional effort required to evaluate our paper.
>
> Best regards,
>
> The Authors

---

### Meta-Review · Area_Chair_J2oW · 2026-01-02

**Summary:**

The reviews praised the clarity of the paper and the timeliness of the contributions. More precisely, they appreciated the release of a dataset with verified reasoning traces, the novel application of reinforcement learning for time-series reasoning and the rigor of the experiments.

**Reviewer Concerns:**

The following issues were solved during the rebuttal:
positioning w.r.t. time-series reasoning literature and comparison with existing models and with established benchmarks, model sensitivity to number of verified trajectories, robustness of results to prompt variation, reward design, output sampling.

**Reviewer Scores:**

All reviewers' scores should have been raised as the issues were sufficiently tackled.

---

### Decision · Program_Chairs · 2026-01-26

Accept (Poster)